# A wake-active locomotion circuit depolarizes a sleep-active neuron to switch on sleep

**Elisabeth Maluck**[1,2☉], **Inka Busack**[1,2☉], **Judith Besseling**[1], **Florentin Masurat**[1], **Michal Turek**[1], **Karl Emanuel Busch**[3], **Henrik Bringmann**[1,2]*

**1** Max Planck Institute for Biophysical Chemistry, Göttingen, Germany, **2** University of Marburg, Marburg, Germany, **3** University of Edinburgh, Edinburgh, United Kingdom

☉ These authors contributed equally to this work.
* henrik.bringmann@biologie.uni-marburg.de

**Data Availability Statement:** All data are available as part of this manuscript. Data used for figure

## Abstract

Sleep-active neurons depolarize during sleep to suppress wakefulness circuits. Wake-active wake-promoting neurons in turn shut down sleep-active neurons, thus forming a bipartite flip-flop switch. However, how sleep is switched on is unclear because it is not known how wakefulness is translated into sleep-active neuron depolarization when the system is set to sleep. Using optogenetics in *Caenorhabditis elegans*, we solved the presynaptic circuit for depolarization of the sleep-active RIS neuron during developmentally regulated sleep, also known as lethargus. Surprisingly, we found that RIS activation requires neurons that have known roles in wakefulness and locomotion behavior. The RIM interneurons—which are active during and can induce reverse locomotion—play a complex role and can act as inhibitors of RIS when they are strongly depolarized and as activators of RIS when they are modestly depolarized. The PVC command interneurons, which are known to promote forward locomotion during wakefulness, act as major activators of RIS. The properties of these locomotion neurons are modulated during lethargus. The RIMs become less excitable. The PVCs become resistant to inhibition and have an increased capacity to activate RIS. Separate activation of neither the PVCs nor the RIMs appears to be sufficient for sleep induction; instead, our data suggest that they act in concert to activate RIS. Forward and reverse circuit activity is normally mutually exclusive. Our data suggest that RIS may be activated at the transition between forward and reverse locomotion states, perhaps when both forward (PVC) and reverse (including RIM) circuit activity overlap. While RIS is not strongly activated outside of lethargus, altered activity of the locomotion interneurons during lethargus favors strong RIS activation and thus sleep. The control of sleep-active neurons by locomotion circuits suggests that sleep control may have evolved from locomotion control. The flip-flop sleep switch in *C. elegans* thus requires an additional component, wake-active sleep-promoting neurons that translate wakefulness into the depolarization of a sleep-active neuron when the worm is sleepy. Wake-active sleep-promoting circuits may also be required for sleep state switching in other animals, including in mammals.

generation are provided in the S1 and S2 Data supplemental Excel sheets.

**Funding:** This work was supported by the Max Planck Society (Max Planck Research Group) to H. B., a European Research Council Starting Grant under the European Union's Horizon 2020 research and innovation program (ID: 637860; SLEEPCONTROL) to H.B., and the University of Marburg to H.B. The funders had no role in study design, data collection and analysis, decision to publish, or preparation of the manuscript.

**Competing interests:** The authors have declared that no competing interests exist.

**Abbreviations:** ΔF/F, change of fluorescence over baseline; APTF-1, Activating enhancer binding Protein 2 Transcription Factor 1; ArchT, archaerhodopsin from *Halorubrum* strain TP009; ATR, all-trans-retinal; dFB, dorsal Fan-shaped Body; DIC, differential interference contrast; EEG, electroencephalogram; FLP-11, FMRF-Like Peptide 11; GCaMP, genetically encoded calcium indicator; HA, histamine; HisCl, Histamine-gated Chloride channel; ICE, Caspase-1/Interleukin-1 converting enzyme; NGM, Nematode Growth Medium; NREM, non-Rapid Eye Movement; PVC, Posterior Ventral Cord neuron of the lumbar ganglion class name; R, fluorescence of GCaMP divided by fluorescence of mKate2; R2, ring neurons of the ellipsoid body; ReaChR, Red-activatable channelrhodopsin; RIM, Ring Interneuron M class name; RIS, Ring Interneuron S class name; SDQL, Posterior lateral interneuron class name—left cell.

# Introduction

Sleep is a behavior that affects many, if not all, physiological processes. Disorders and curtailment of sleep affect the lives of 10% to 30% of the adult population of modern societies. Sleep loss is associated with an increased risk of infection [1], cardiovascular disease [1], psychiatric disease (including depression [2,3]), obesity [4,5], type 2 diabetes [4,5], and cancer [1]. The high prevalence of insomnia and insufficient sleep quality thus presents a massive unmet health and economic problem [1,3–5]. To understand how sleep behavior is generated, it is crucial to solve the underlying neural circuits.

Sleep circuits require inhibitory sleep-active sleep-promoting neurons, which depolarize specifically at sleep onset and actively induce sleep by releasing inhibitory neurotransmitters, GABA and neuropeptides, to dampen arousal and the activity of wake circuits [6]. Sleep behavior induced by inhibitory sleep-active neurons includes the suppression of voluntary movements and sensory perception, reversibility, and homeostasis [7]. Inhibitory sleep-active neurons suppress wake circuits and can be rapidly suppressed by arousing stimulation to allow for quick awakening. Forced wakefulness is followed by an increase of sleep-active neuron depolarization, which leads to homeostatic sleep corrections. Thus, understanding sleep control requires comprehension of the circuit mechanisms that determine when and how much inhibitory sleep-active neurons depolarize [6,8].

Circuits control the depolarization of inhibitory sleep-active neurons. For example, wake-active wake-promoting neurons promote arousal and suppress inhibitory sleep-active neurons, whereas sleep need causes sleep-active neuron depolarization. Thus, sleep-active sleep-promoting and wake-active wake-promoting neurons form a flip-flop switch, which ensures that sleep and wake exist as discrete states. This sleep switch is under the control of arousal that favors wake and inhibits sleep through the suppression of sleep-active neurons by inhibitory wake-active neurons [6,9]. It has been proposed that sleep induction is favored by disinhibition of inhibitory sleep-active neurons [10–12]; also, excitatory sleep-active neurons exist that might perhaps present activators of inhibitory sleep-active neurons [13]. However, the forces and mechanisms that flip the sleep switch from wake to sleep when an organism gets sleepy cannot be satisfactorily explained by the present circuit models as it is unclear how sleep-active neurons are turned on when the system is set to sleep.

Sleep is under circadian and homeostatic controls that determine the timing of sleep and ensure that enough of this essential physiological state takes place [14]. Sleep homeostasis comprises multiple mechanisms that act on different timescales. On long timescales, sleep is a function of prior wakefulness, i.e., prolonged wakefulness leads to increased sleep propensity, and sleep loss triggers compensatory increases in the intensity or duration of sleep. This chronic sleep homeostasis likely is mediated by several parallel mechanisms. For example, in mammals, somnogens such as adenosine accumulate during wakefulness, leading to the inhibition of wake-promoting neurons [15,16]. In *Drosophila*, activity-dependent plasticity of sleep-promoting neurons increases during wakefulness to increase subsequent sleep [17,18]. On short timescales, acute homeostasis determines whether the system's actual state matches the system's set point and carries out corrective action if those values do not match. For example, to homeostatically maintain sleep despite disturbance, micro-arousals need to be compensated for. In humans, homeostatic sleep maintenance can be seen in electroencephalogram (EEG) recordings in the form of k-complexes, in which a spontaneous or evoked short cortical up state is followed by a down state [19–21]. Homeostatic sleep maintenance is also found during sleep in *C. elegans*, in which sleep bouts are interrupted by short motion bouts, with the length of a motion bout correlating with the length of the subsequent sleep bout [22,23]. Thus, across systems, homeostatic sleep maintenance requires constant surveillance of sleep and corrective action.

Sleep-active sleep-promoting neurons are conserved regulators of sleep and have been found both in vertebrates as well as in invertebrates [8,24]. Mammals possess several populations of sleep-active neurons, most of which are inhibitory, across the brain. These neurons reside in the anterior hypothalamus, brain stem, and cortex [6,12]. Excitatory sleep-active neurons were found in the periocular midbrain that project to inhibitory sleep-active neurons in the anterior hypothalamus, the role of which could be to activate inhibitory sleep-active neurons [13]. Studying sleep in less complex brains facilitates sleep circuit analysis. In *Drosophila*, sleep-promoting neurons are found at several locations in the brain. A well-characterized population of sleep-promoting neurons is formed by neurons residing in the dorsal Fan-shaped Body (dFB). R2 ring neurons of the ellipsoid body accumulate homeostatic sleep pressure over time to promote activation of sleep-promoting dFB neurons, probably by an indirect mechanism [17,18]. *C. elegans* possesses a single inhibitory sleep-active neuron called RIS. Like its mammalian counterparts, RIS depolarizes at sleep onset. RIS is crucial for sleep induction because its ablation leads to a virtually complete loss of detectable sleep bouts [25–27]. The small, invariant nervous system, its mapped connectome, and the transparency of *C. elegans* facilitate neural circuit analysis [28]. However, the specific neural circuits that control RIS activity are not yet understood.

*C. elegans* shows sleep behavior during many stages and conditions. Here, we analyzed sleep behavior during development, also known as lethargus, the stage prior to each of the 4 molts during larval development [8,27,29–31]. We used optogenetics to dissect the neural circuits that control the activation of the sleep-active RIS neuron in *C. elegans*. We found a third and novel important element of the flip-flop switch: interneurons that are active during wakefulness and that are known to control locomotion are required for RIS activation and sleep. These findings suggest a tripartite flip-flop circuit model that can explain how arousing stimulation inhibits RIS depolarization, how RIS depolarization is homeostatically controlled, and how reduced arousal can induce RIS depolarization. Our RIS circuit model has 2 important implications for understanding sleep control: (1) it suggests that sleep control has evolved from circuits controlling locomotion; and (2) sleep induction requires an important third element, wake-active sleep-promoting neurons, which translate wakefulness into sleep when the animal is sleepy but awake.

## Results

### Interneurons known to govern locomotion behavior control RIS activity

RIS is crucially required for sleep and typically activates during sleep bouts (Fig 1A) [25]. However, the presynaptic driver neurons that activate and control this neuron are not known. To identify the circuits controlling RIS activation, we optogenetically tested the role of neurons that are presynaptic to RIS according to the *C. elegans* connectome [28]. The neurons called AVJL, CEPDL, URYVL, RIMR, PVCL, and SDQL have been shown to be presynaptic to RIS [28,32]. To find out how these presynaptic neurons control RIS, we activated them with ReaChR (red-activatable channelrhodopsin) and green light and followed RIS calcium activity using GCaMP (a genetically encoded calcium indicator) during and outside of lethargus. We confirmed the expression of ReaChR through a fused fluorescent reporter (mKate2). AVJ, CEPD, URYV, RIM, PVC, and SDQ each are a pair of 2 neurons, of which only one is presynaptic to RIS. Because only promoters that express in both neurons of each pair are available—and because the 2 neurons of each pair are in close proximity—we always manipulated both neurons of the neuronal pair (except for SDQL) [28,32]. Because there were no specific promotors available for the expression in SDQL and PVC, we expressed ReaChR using semi-specific promoters and selectively illuminated only the presynaptic neuron class. We used L1

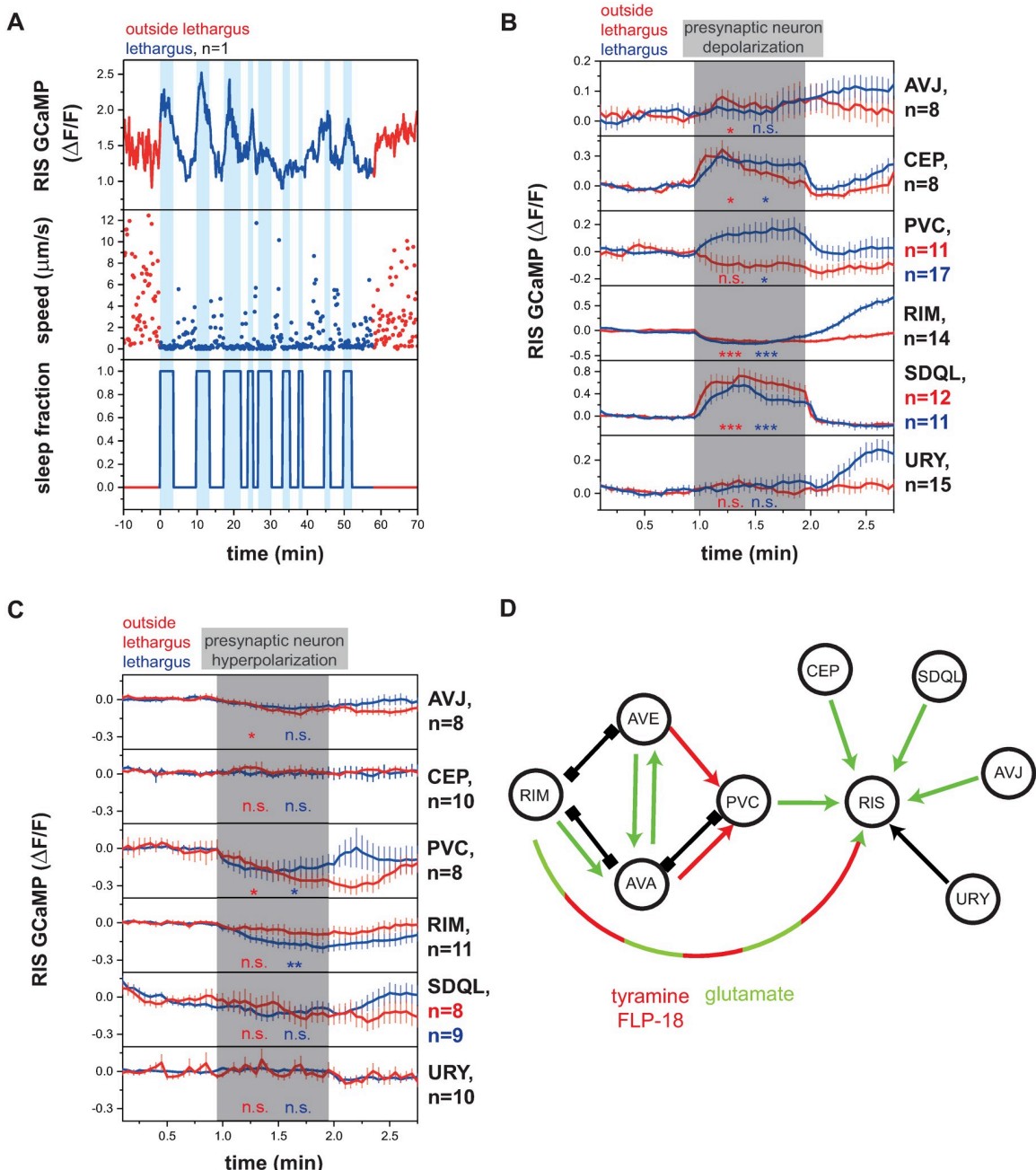

**Fig 1. Presynaptic neurons control the activity of the sleep-active RIS neuron.** (A) Sample trace of RIS activity and worm locomotion behavior outside of and during lethargus. RIS has no strong calcium transients outside of lethargus but shows strong activity transients during lethargus. Upon RIS activation, worms enter sleep bouts. (S1 Data, Sheet 1A). (B) Presynaptic neurons activate or inhibit RIS outside of and during lethargus. For statistical calculations, neural activities before the stimulation period (0–0.95 min) were compared to activity levels during the stimulation period (1–1.95 min). $^*p < 0.05$, $^{**}p < 0.01$, $^{***}p < 0.001$, Wilcoxon signed rank test. (S1 Data, Sheet 1B). (C) RIS activity decreases upon optogenetic PVC and RIM hyperpolarization. Statistical calculations were performed as described in panel B, but in experiments in which SDQL was stimulated, baseline activity levels were calculated over the time interval from 0.6 to 0.95 min. Baseline activity levels were calculated starting from 0.6 min as baseline activity levels were instable before that time point. $^*p < 0.05$, $^{**}p < 0.01$, Wilcoxon signed rank test. (S1 Data, Sheet 1C). (D) Circuit model of the RIS presynaptic regulatory network. Activating synaptic input is shown as green arrows, inhibitory synaptic input is shown as red arrows, and unclear synaptic input is shown as black arrow. Gap junctions are indicated as black connections. Neurons that are presynaptic to RIS present mostly activators. PVC is essential for lethargus-specific RIS activation. RIM can inhibit RIS through tyramine and FLP-18 and can activate RIS with glutamate. ΔF/F, change of fluorescence over baseline; FLP-18, FMRF-Like Peptide 18; GCaMP, genetically encoded calcium indicator; n.s., not significant; PVC, Posterior Ventral cord neuron class name; RIM, Ring Interneuron M class name; RIS, Ring Interneuron S class name; SDQL, Posterior lateral interneuron class name—left cell.

larvae for most of the optogenetic experiments to dissect the circuit. As SDQL is born postem-bryonically and likely is not yet functional during the L1 stage, we used L4 larvae to assay its function [33]. We compared the effects of optogenetic stimulation outside and during lethar-gus, defined as the period prior to the molt during which the animals do not feed [34]. Before lethargus, we measured an activation of RIS upon depolarization of AVJ, CEP, and SDQL. During lethargus, the activation of CEP, PVC, and SDQL caused RIS activation (Fig 1B and S1A Fig).

All neurons showed consistent effects on RIS depolarization except RIM. RIM is known to play complex roles in controlling behavior and is involved in seemingly opposing behaviors. For example, specific RIM activation can trigger a reversal [35], whereas RIM inhibition has been suggested to be required for reversals through an alternative circuit [36]. We performed optogenetic depolarization experiments of RIM expressing ReaChR using 2 different promot-ers, the *tdc-1* promoter, which is known to express strongly, and the *gcy-13* promoter, which is known to express at a lower level [37]. Activation of RIM with channelrhodopsin expressed from the *tdc-1* promoter has previously been shown to cause reversals [35], and we observed that activation of RIM using ReaChR expressed from this promoter led to RIS inhibition (Fig 1B, RIM panel). The *tdc-1* promoter expresses strongly in RIM, but also weakly in RIC [38]. To test whether the inhibitory effect of *tdc-1* promoter-driven ReaChR expression on RIS was caused by RIC, we also specifically expressed ReaChR in RIC using the *tbh-1* promoter [38]. Specific RIC activation led to RIS activation rather than inhibition (S1B Fig). Therefore, the *tdc-1*::*ReaChR*-mediated RIS inhibition appears to stem from RIM activation. Activating RIM with the weaker *gcy-13* promoter did not cause any net effects on RIS when all trials were aver-aged (S1C Fig). Visual inspection of the individual trials, however, showed that RIM activation could either inhibit or activate RIS. We therefore sorted single trials for the *gcy-13* experiment into 2 classes, in which RIM either activated or inhibited RIS function (S1D Fig). The activa-tion or inhibition of RIS by RIM was indistinguishable during the beginning or end of lethar-gus (S1E Fig).

To confirm that RIM can both activate and inhibit RIS, we tested whether activation and inhibition are mediated by different neurotransmitters. We tested the effects of RIM activation on RIS in mutants, which lack transmitters that are known to be expressed in RIM. The RIM neurons are well known to inhibit downstream neurons using tyramine, which requires the *tdc-1* gene [38], and also express neuropeptides (FMRF-Like Peptide 18 [FLP-18]) encoded by the *flp-18* gene [39]. To test whether RIM can inhibit RIS using these known transmitters, we analyzed mutant worms that lack functional *flp-18* and *tdc-1*. Individual inactivation of *flp-18* and *tdc-1* reduced—and double mutation abolished—the inhibition of RIS by RIM (S2 Fig). Therefore, the transmitters tyramine and FLP-18 are together responsible for RIS inhibition by RIM. We next tested activation of RIS by RIM in *eat-4(ky5)* mutant larvae, which lack gluta-matergic signaling in many neurons, including RIM [40,41]. RIS activation by RIM activation was completely gone in *eat-4(ky5)* mutant larvae (S3 Fig, we used L4 larvae for this assay as the response was more robust). Therefore, glutamate is required for RIS activation by RIM. Together, these results suggest that RIM can act both as an activator as well as an inhibitor of RIS by employing different neurotransmitters, with weaker activation allowing for RIS activa-tion and stronger activation favoring inhibition.

The majority of synaptic inputs into RIS that we studied had activating effects; the sole inhibitory effect was observed after strong activation of RIM, whereas weaker RIM activation could also lead to RIS activation. The CEP, URY, and SDQL neurons present sensory receptors and might play a role in activating RIS in response to stimulation. For example, CEP might activate RIS as part of the basal slowing response [42,43]. The PVCs appeared to be strong acti-vators of RIS specifically during lethargus. This suggests either that the PVC-to-RIS

connection might be specific to lethargus or that it has not yet matured during the mid-L1 stage. We therefore repeated the experiment and activated PVC in L2 larvae. PVC activated RIS both during and outside of lethargus, but the activation during lethargus was much stronger, suggesting that the activation of RIS by PVC is strongly enhanced during lethargus (S4 Fig).

To find out which presynaptic neurons are required for inhibition or activation of RIS during lethargus, we tested the effect of optogenetic inhibition of the presynaptic neurons on RIS activation. We used ArchT (archaerhodopsin from *Halorubrum* strain TP009), which hyperpolarizes neurons by pumping protons out of the cell [44,45]. As earlier, we verified the expression of ArchT in neurons of interest by using an mKate2-tagged version. As in the ReaChR experiments, we specifically illuminated each presynaptic neuron class and quantified RIS activation using calcium imaging. Before lethargus, inhibition of AVJ and PVC led to an inhibition of RIS, whereas inhibition of the other neurons tested had no acute statistically significant effect on RIS (optogenetic RIM hyperpolarization using the stronger *tdc-1* promoter in worms outside of lethargus showed a tendency to inhibit RIS function [$p = 0.0539$; $N = 11$ animals], whereas the weaker *gcy-13* promoter had no detectable effect). During lethargus, optogenetic inhibition of PVC and RIM (using the stronger *tdc-1* promoter) led to significant RIS inhibition, whereas there was no effect seen for the other neurons (Fig 1C and S5A Fig; inhibition of RIM using the weaker *gcy-13* promoter only produced a tendency but no statistically significant net effect, S5B and S5C Fig).

Absence of an effect of optogenetic inhibition of presynaptic neurons could mean either that these neurons are not required for RIS activation, that the inhibition was not strong enough, or that they may act redundantly (we did not find any evidence for redundancy, at least for CEP and URY, S5D Fig). Our optogenetic analysis revealed a complex set of presynaptic inputs for regulation of RIS activity (Fig 1D). The optogenetic depolarization experiments suggest that CEP, PVC, RIM, and SDQL present the most potent presynaptic activators of RIS. The capacity of PVC to activate RIS is strongly increased during lethargus, indicating that this neuron is involved in the lethargus-specific activation of RIS. The optogenetic hyperpolarization experiments suggest that PVC and RIM are essential presynaptic activators of RIS during lethargus. Therefore, we focused our analysis on PVC and RIM neurons.

## PVC becomes resistant to inhibition during lethargus

Neuronal activation and silencing experiments revealed PVC as a main activator of RIS. These results predict that neuronal activity of PVC should correlate with RIS activation and sleep bouts. To test for such correlation, we measured the activity of both neurons simultaneously. Because the calcium transients observable in PVC are typically small [46] and could not be detected in our assays in mobile worms (data not shown), we immobilized the larvae and used RIS activation as a proxy for sleep bouts. We extracted both RIS and PVC activity and aligned all data to the RIS activation maxima. This analysis showed that PVC activated approximately 1 min earlier than RIS and reached its maximum activation approximately 1.5 min earlier than RIS. PVC activity decreased slowly during the RIS transient (Fig 2A). This result is consistent with a role for PVC in promoting RIS depolarization.

PVC inhibition reduced RIS activity in immobilized animals, but it is unclear how PVC inhibition affects behavior. To be able to test the effects of PVC inhibition on behavior without affecting the other command interneurons, we chose a more specific promoter for expression in PVC from single-cell RNA sequencing data. There was no gene in the available datasets that was expressed only in the cluster of cells containing PVC, but the previously uncharacterized gene *zk673.11* was expressed specifically in PVC and in only a few other neurons excluding

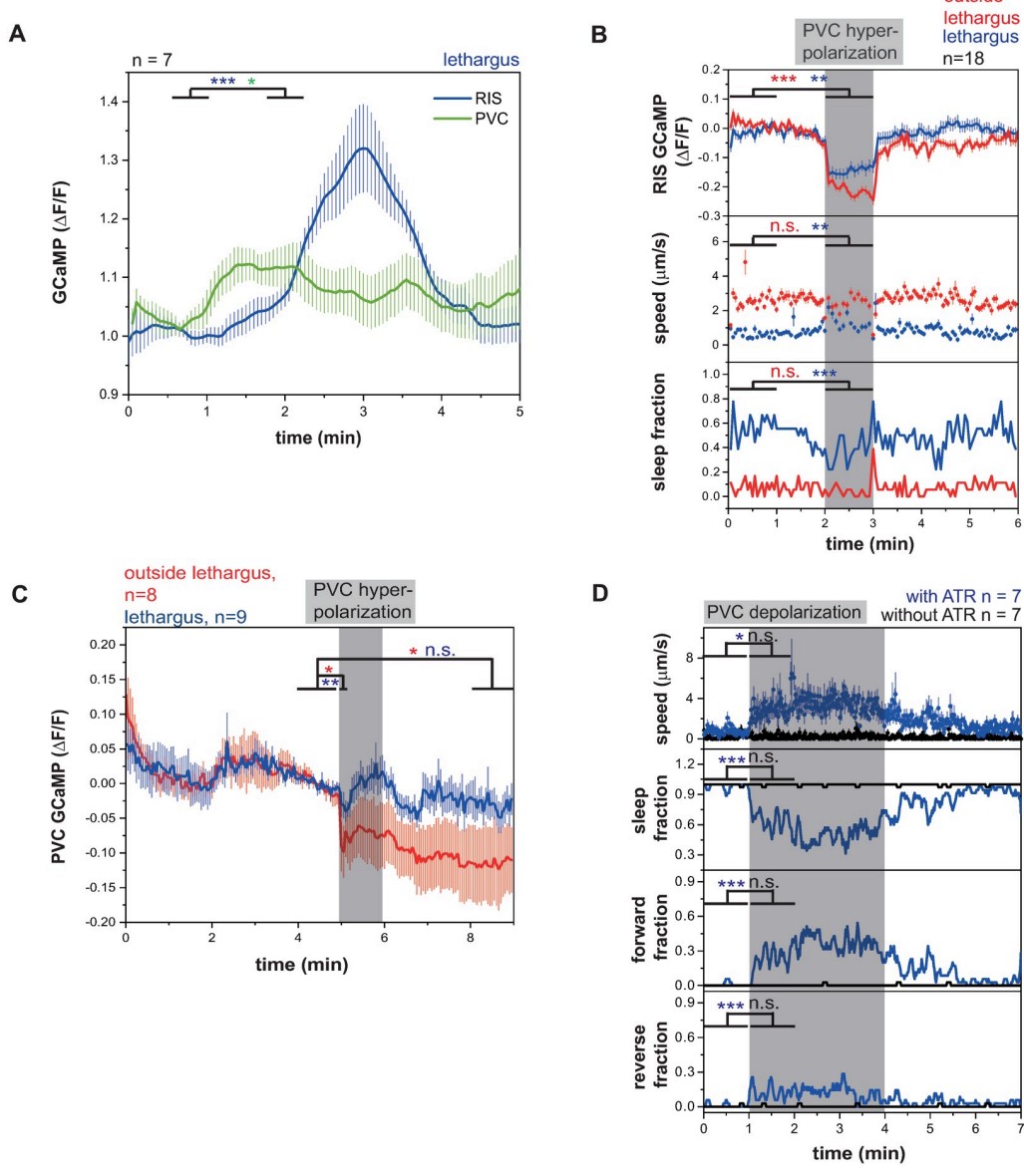

**Fig 2. PVC is an RIS activator that becomes resistant to inhibition during lethargus, but PVC activation is not sufficient for sleep induction.** (A) Simultaneous PVC and RIS GCaMP traces aligned to RIS peaks in fixed L1 lethargus worms. PVC activates before the RIS peak and stays active until the peak. $^{*}p < 0.05$, $^{**}p < 0.01$, Wilcoxon signed rank test. (S1 Data, Sheet 2A). (B) PVC hyperpolarization inactivates RIS and induces behavioral activity. PVC hyperpolarization was performed by expressing ArchT under the *zk637.11* promoter. In contrast to the *nmr-1* promoter, the *zk637.11* promoter lacks expression in head command interneurons. $^{**}p < 0.01$, $^{***}p < 0.001$, Wilcoxon signed rank test for GCaMP and speed, Fisher's exact test for sleep fraction. (S1 Data, Sheet 2B). (C) During lethargus, PVC becomes resistant to inhibition. Outside of lethargus, its inhibition is stronger and continues beyond the end of optogenetic stimulation. During lethargus, PVC activity levels return back to baseline already during the stimulation period. $^{*}p < 0.05$, $^{**}p < 0.01$, Wilcoxon signed rank test. (S1 Data, Sheet 2C). (D) PVC activation translates into mostly a forward mobilization in L1 lethargus. $^{*}p < 0.05$, $^{***}p < 0.001$, Wilcoxon signed rank test for Speed. Fisher's exact test for fraction of direction. (S1 Data, Sheet 2D). ArchT, archaerhodopsin from *Halorubrum* strain TP009; ATR, all-trans-retinal; ΔF/F, change of fluorescence over baseline; GCaMP, genetically encoded calcium indicator; n.s., not significant; PVC, Posterior Ventral cord neuron class name; RIS, Ring Interneuron S class name.

other command interneurons [47,48] (personal communication from J. Packard to H. Bring-mann; S6 Fig). Hyperpolarization of PVC using ArchT driven by the *zk673.11* promoter led to an acute inhibition of RIS, an increase in locomotion, and a reduction of sleep (Fig 2B). Hyper-polarization of PVC using ArchT also strongly inhibited RIS outside of lethargus. This experiment confirmed the role of the PVCs in activating RIS.

Hyperpolarization of PVC outside of lethargus appeared to have a stronger and longer-lasting effect on RIS inhibition compared with during lethargus (Figs 1C and 2B). This is surprising because PVC is a stronger activator of RIS during lethargus in the optogenetic activation experiments (Fig 1B). This effect could be explained if PVC responded more severely to inhibition outside of lethargus. We tested this idea by inhibiting PVC using ArchT and green light and simultaneously imaged PVC activity. PVC hyperpolarization was stronger in worms outside of lethargus, and PVC remained inhibited after the optogenetic manipulation. During lethargus, PVC was only weakly inhibited at the beginning of optogenetic stimulation and returned to baseline levels already during the stimulation (Fig 2C). We also tested whether optogenetic excitability of PVC was modulated during lethargus but could not find any differences in excitability of PVC during or outside of lethargus (S7A Fig). Thus, PVC is more susceptible to inhibition outside of lethargus but becomes resistant to inhibition during lethargus. This effect can explain the stronger hyperpolarization of RIS during PVC inhibition outside of lethargus, and this effect likely presents an important modulation of the circuit to favor PVC activation and thus RIS activation during lethargus.

PVC is known to promote forward movement upon posterior mechanical stimulation, and optogenetic stimulation of PVC in adults has been shown to promote forward locomotion [49,50]. Our data showed that PVC also activates the RIS neuron, and consistent with this observation, mechanical stimulation caused RIS activation (S7B Fig). This suggests that PVC activates RIS to modulate forward locomotion speed and to promote sleep. However, it is unclear how PVC can promote forward motion and sleep, as these are two seemingly opposing behaviors. We therefore tested whether optogenetic stimulation of PVC in larvae induces sleep behavior. We activated PVC using *nmr-1*::*ReaChR* in mobile L1 larvae during lethargus and specifically illuminated the tail of the animal, which contains the cell bodies of the PVC neurons but not the other *nmr-1*-expressing neurons. We quantified the speed as well as the direction of movement of the worm. During PVC activation during lethargus, the worms visibly accelerated movement and mostly crawled forward, but we could not see induction of sleep behavior during optogenetic stimulation (Fig 2D). Consistent with this finding, optogenetic PVC activation during and before lethargus always led to the activation of AVB interneurons, which are known to be premotor neurons required for forward locomotion [49] (S7C and S7D Fig). Together, these experiments showed that PVC activates prior to RIS and is required for RIS activation. However, its activation alone does not seem to be sufficient to induce sleep behavior.

## RIS and PVC activate each other forming a positive feedback loop

PVC presents a major activator of RIS, but how a forward command interneuron can cause strong and state-specific activation of the RIS neuron during sleep bouts is not clear. We therefore tested how optogenetic RIS activation affects PVC activity. We selectively activated RIS using ReaChR and measured calcium activity in PVC in immobilized animals. Upon RIS stimulation, PVC immediately displayed unexpectedly strong calcium transients, which were slightly stronger during lethargus (Fig 3A and S8A Fig). These results show that PVC and RIS activate each other, thus forming a positive feedback loop. The sleep-inducing RIS neuron has so far only been shown to inhibit other neurons, making PVC the first neuron that is not

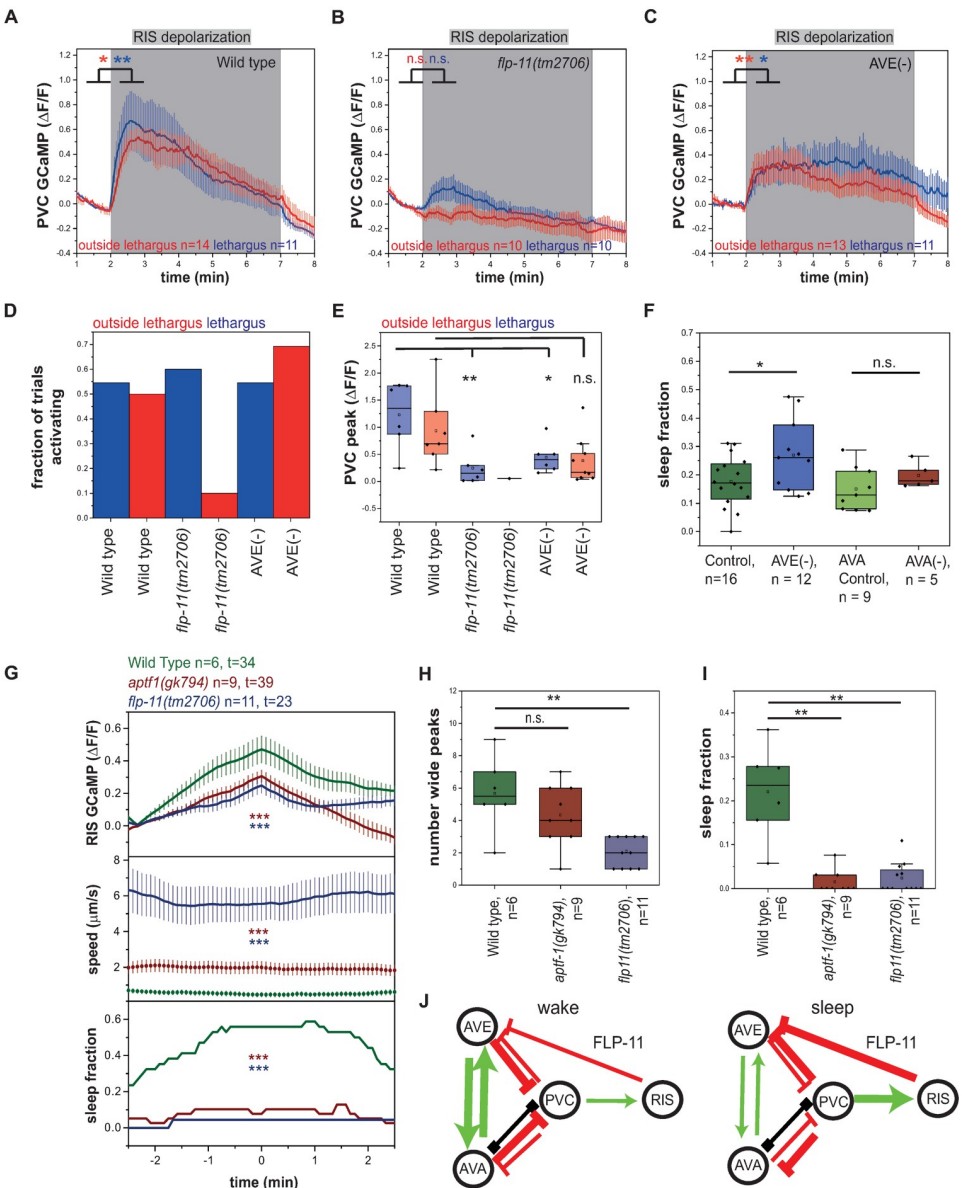

**Fig 3. RIS and PVC activate each other, forming a positive feedback loop.** (A–E) RIS depolarization leads to a strong PVC depolarization outside of and during lethargus. This PVC depolarization is almost abolished in *flp-11 (tm2706)*, and it is significantly reduced in AVE-ablated worms. $^*p < 0.05$, $^{**}p < 0.01$, Wilcoxon signed rank test (S1 Data, Sheets 3A, 3B, 3C-E). (F) AVE-ablated worms show increased sleep. AVA-ablated worms do not show a significant sleep phenotype. Shown are sleep fractions during lethargus. $^*p < 0.05$, Kolmogorov-Smirnov test (S1 Data, Sheet 3F). (G) RIS does not reach the same activation levels in *aptf-1(gk794)* and *flp-11(tm2706)* mutants compared to wild-type worms. *aptf-1(gk794)* and *flp-11(tm2706)* mutants neither immobilize nor sleep during RIS activation. $^{***}p < 0.001$, Welch test (S1 Data, Sheet 3G-I). (H) *flp-11(tm2706)* mutants have significantly fewer wide RIS peaks. *aptf-1(gk794)* mutants display the same amount of wide RIS peaks as wild-type worms. $^{**}p < 0.01$, Kolmogorov-Smirnov test (S1 Data, Sheet 3G-I). (I) *flp-11(tm2706)* and *aptf-1(gk794)* mutants do not show sleep during lethargus. $^{**}p < 0.01$, Kolmogorov-Smirnov test (S1 Data, Sheet 3G-I). (J) A circuit model for the positive feedback loop between RIS and PVC. Activating synaptic input is shown as green arrows, inhibitory synaptic input is shown as red arrows, and gap junctions are indicated as black connections. During wakefulness, reverse command interneurons inhibit PVC so that PVC does not activate RIS. During lethargus, PVC directly activates RIS, which then inhibits reverse command interneurons through FLP-11. This may speculatively disinhibit PVC, leading to a positive feedback. ΔF/F, change of fluorescence over baseline; FLP-11, FMRF-Like Peptide 11; GCaMP, genetically encoded calcium indicator; n.s., not significant.

inhibited but is activated by RIS. For example, command interneurons such as AVE and AVA and other neurons are not activated but are inhibited by RIS [25].

RIS induces sleep through the release of neuropeptides with the major sleep-inducing neuropeptides encoded by the *flp-11* gene [51]. To test whether FLP-11 neuropeptides are required for RIS-induced PVC activation, we repeated the optogenetic RIS activation with simultaneous PVC calcium measurement in an *flp-11* deletion mutant. RIS-induced PVC activation was almost completely abolished in the *flp-11* deletion (reduction of transient maximum by 79% during lethargus), indicating that FLP-11 neuropeptides are required for RIS-induced PVC activation (Fig 3B).

While PVC is presynaptic to RIS, RIS is not presynaptic to PVC [28,32]. The activation of PVC by RIS could involve diffusional mechanisms or could be indirect through other neurons, perhaps mediated by the inhibition of a PVC inhibitor such as AVA/AVD/AVE. RIS has been shown to inhibit AVA/AVE [25], and RIS is presynaptic to AVE [28,32], suggesting that PVC activation involves inhibition of AVE. We therefore repeated RIS activation and PVC calcium imaging in a strain in which AVE was impaired through expression of tetanus toxin [52]. The initial PVC activation maximum after AVE impairment was reduced by 43% during lethargus, but subsequent PVC activity was increased (Fig 3C–3E). AVE is connected to other reverse command interneurons, which collectively inhibit PVC [28,53]. This circuit design suggests that AVE might play a dual role in controlling RIS activity. It should have a positive role in mediating activation of PVC through RIS and thus could promote the feedback loop, but it should also have an inhibiting role by promoting PVC inhibition. To test for a role of the arousal neurons AVE and AVA in sleep, we inhibited AVE with tetanus toxin [52] and AVA using HisCl (Histamine-gated Chloride channel) [54] and quantified sleep amount. Whereas we could not find any effect of AVA impairment on sleep amount, AVE impairment led to an average increase of sleep by 42% (Fig 3F). Together, these data suggest that PVC and RIS rely on positive feedback for their activation that involves the release of FLP-11 neuropeptides and inhibition of PVC by AVE.

If depolarization of RIS activates PVC, what consequences does hyperpolarization of RIS have on PVC activity? To answer this question, we measured the response of PVC to RIS inhibition. We hyperpolarized RIS optogenetically for 1 min using ArchT and measured the activity of PVC. Interestingly, PVC showed a small but significant activity increase during RIS inhibition, an effect that was increased during lethargus (S8B Fig). The disinhibition of PVC by RIS inactivation is likely not direct and may reflect a general increase in neuronal and behavioral activity that is caused by RIS inhibition and that extends to the PVC neurons. Because PVC is a major activator of RIS, its disinhibition could be part of a homeostatic feedback regulation.

Our results suggest that there is a positive feedback from sleep induction onto RIS activation and that full RIS activation is only possible when sleep is successfully induced, explaining the strong correlation of RIS depolarization and sleep-bout induction [27]. This model would predict that RIS transients are dampened if RIS is not able to induce sleep bouts. To test this idea, we analyzed RIS calcium transients in *aptf-1(−)* mutant worms in which RIS still shows depolarization transients during lethargus but cannot efficiently induce quiescence [25,51]. In *aptf-1(−)* mutant animals, calcium transient maxima were reduced by about 35% (Fig 3G–3I). A major function of APTF-1 (Activating enhancer binding Protein 2 Transcription Factor 1) is the expression of FLP-11 neuropeptides that are required for quiescence induction [51]. To test whether FLP-11 neuropeptides play an essential role in shaping RIS transients, we measured RIS calcium transients in mutant worms carrying a deletion of *flp-11*. These mutant animals showed only a reduced number of long RIS transients that were of reduced size (Fig 3G–3I). *flp-11(−)* showed, however, many short RIS transients (S8C–S8F Fig) that were not

associated with sleep bouts but may reflect attempts to induce sleep bouts. These results are consistent with the idea that sleep induction is a self-enforcing process in which RIS-mediated inhibition of brain activity through FLP-11 neuropeptides promotes long RIS calcium transients (Fig 3J).

We next tested what feedback interaction exists between RIM and RIS neurons. We optogenetically depolarized or hyperpolarized RIS and measured RIM activity. RIS activation did not significantly change RIM activity, but there was a small inhibitory trend (S8G Fig). RIS inhibition led to an activation of RIM (S8H Fig). These results show that, while RIM can activate as well as inhibit RIS, RIS is an inhibitor of RIM.

## RIM can activate RIS, but its activation is not sufficient for sleep induction

A second important activator of RIS is RIM. We therefore asked whether RIM, similar to PVC, also is active prior to RIS depolarization and sleep bouts. We measured RIM activity by imaging GCaMP in moving worms. All sleep bouts were extracted, and RIM activity was aligned to sleep-bout onset. Averaged RIM activity peaked approximately 30 s before the beginning of the sleep bout (Fig 4A). This finding is consistent with a function for RIM in RIS activation. We then asked whether RIM is required for sleep induction. We ablated RIM through expression of *egl-1* under the *tdc-1* promoter. We quantified lethargus sleep in RIM-ablated worms. RIM-ablated larvae showed a normal fraction of sleep, a slightly increased frequency of sleep bouts, and a normal length of sleep bouts (Fig 4B–4D). In analogy to the PVC experiments, we analyzed the effect of optogenetic RIM depolarization on behavior. We first tested behavior caused by activation of RIM with ReaChR driven by the strong *tdc-1* promoter on the locomotion of worms. Consistent with previous findings [35] and our observation that RIS is inhibited under these conditions, RIM activation during lethargus caused mobilization, and larvae crawled mostly backwards (Fig 4E). We next tested for the effects of weaker RIM activation using the *gcy-13* promoter. Activation of RIM caused increased mobility when RIS was inhibited. In trials in which RIM activation led to RIS activation, there was no significant change of speed of the worms (S1D Fig). We next wanted to test whether excitability of RIM is altered during the lethargus state. We therefore activated RIM strongly using the *tdc-1* promoter and measured RIM activity. Outside of lethargus, RIM was strongly excited. During lethargus, however, excitability was strongly reduced (Fig 4F and 4G). In summary, RIM activation is not sufficient to induce sleep. RIM could, however, contribute to strong RIS activation and sleep induction by acting in concert with other neurons. Reduced excitability of RIM during lethargus could favor the activating effect of RIM on RIS while dampening the inhibiting effects of RIM on RIS.

## Interneurons regulating locomotion act in concert to activate RIS

Separate activation of PVC or RIM neurons caused moderate RIS activation but not the strong activation of RIS that is typically associated with sleep bouts. Thus, hypothetically, multiple neurons act in concert to cause strong RIS activation. Our earlier presynaptic neuron analysis suggests that this hypothetical set of neurons should include PVC and RIM interneurons but could also include additional neurons. Our analysis of RIM and PVC points to neurons of the command interneuron circuit for RIS activation, and thus we tested the effects of ablation of a large fraction of the interneurons controlling locomotion. The *nmr-1* promotor expresses in AVA, AVE, AVD, and PVC command interneurons as well as in second-layer RIM neurons [55]. We used a strain that ablates these locomotion-controlling interneurons by expressing the pro-apoptosis regulator ICE (Caspase-1/Interleukin-1 converting enzyme) from the *nmr-1* promotor [55] and measured sleep and RIS activation. Command interneuron ablation

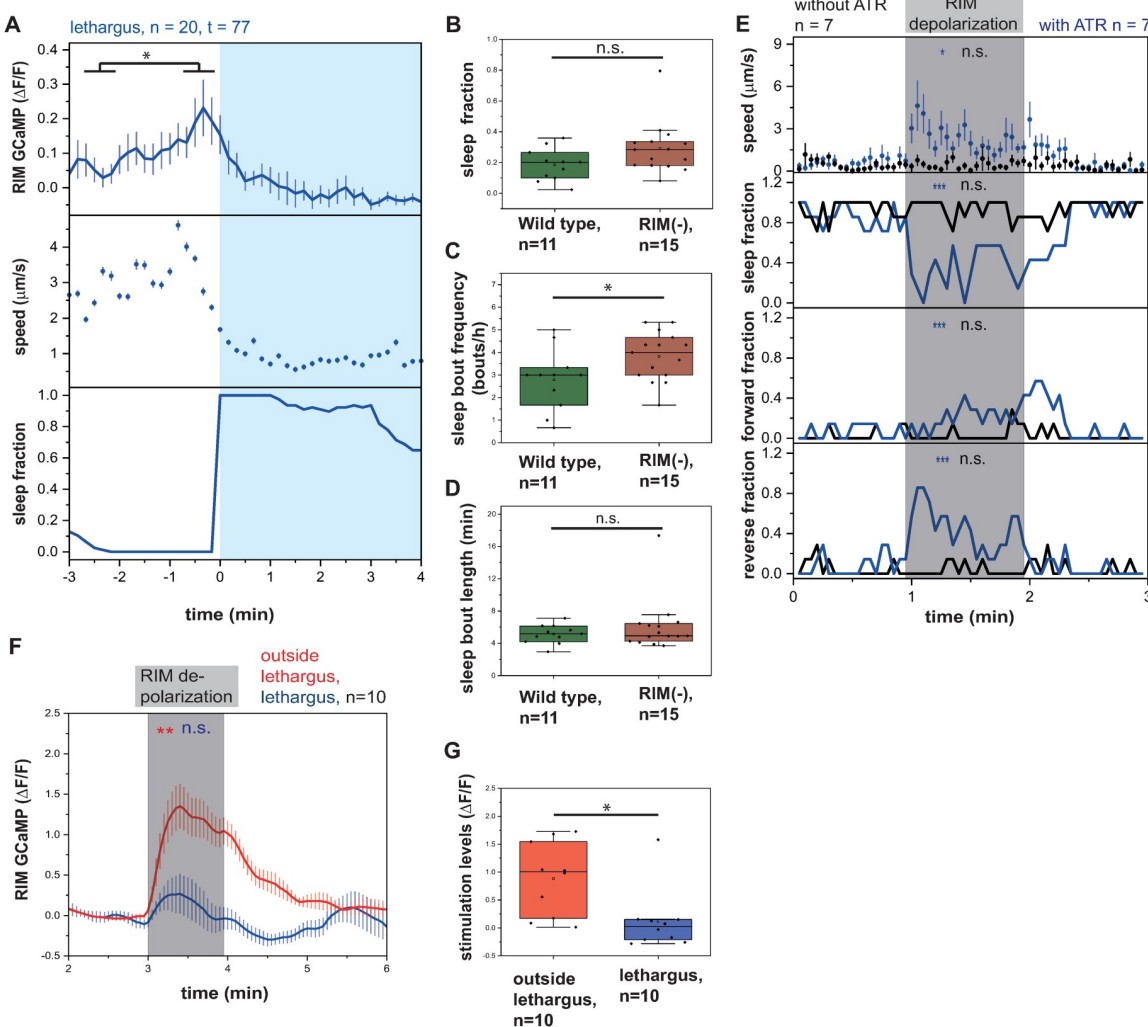

**Fig 4. RIM activity peaks prior to sleep bouts, but RIM activation is not sufficient for sleep induction.** (A) RIM activates prior to sleep bouts. $^*p < 0.05$, Wilcoxon signed rank test (S1 Data, Sheet 4A). (B–D) RIM-ablated worms have an increased sleep-bout frequency, while the sleep fraction and bout duration are not significantly changed during L1 lethargus. RIM was genetically ablated by expressing *egl-1* under the *tdc-1* promoter. $^*p < 0.05$, Kolmogorov-Smirnov test (S1 Data, Sheet 4B-D). (E) RIM depolarization leads to increased mobility and reverse motion. $^*p < 0.05$, $^{***}p < 0.001$, Wilcoxon signed rank test for speed. Fisher's exact test for fraction of direction (S1 Data, Sheet 4E). (F–G) During lethargus, RIM becomes resistant to activation. RIM was optogenetically activated using ReaChR expressed under the *tdc-1* promoter. Outside of lethargus, its activation is stronger (F). Activity levels during the stimulation period were quantified by subtracting baseline activity levels from levels during the stimulation period (G). $^*p < 0.05$, $^{**}p < 0.01$, Wilcoxon signed rank test for GCaMP and Kolmogorov-Smirnov test for quantification of stimulation levels (S1 Data, Sheet 4F-G). ATR, all-trans-retinal; ΔF/F, change of fluorescence over baseline; GCaMP, genetically encoded calcium indicator; n.s., not significant; ReaChR, red-activatable channelrhodopsin; RIM, Ring Interneuron M class name.

reduced sleep bouts during lethargus by about 76% (Fig 5A), and RIS activation was reduced by 63% (S9A Fig). The movement of command interneuron-ablated worms also was slower (S9B Fig). Quiescence bouts did not occur at the beginning of the lethargus phase as defined by cessation of feeding and were only observed around the middle of the lethargus phase (S9C Fig). An independently generated strain that ablates command interneurons using *egl-1* expression—also by using the *nmr-1* promoter—caused a reduction of sleep by 81% (Fig 5A).

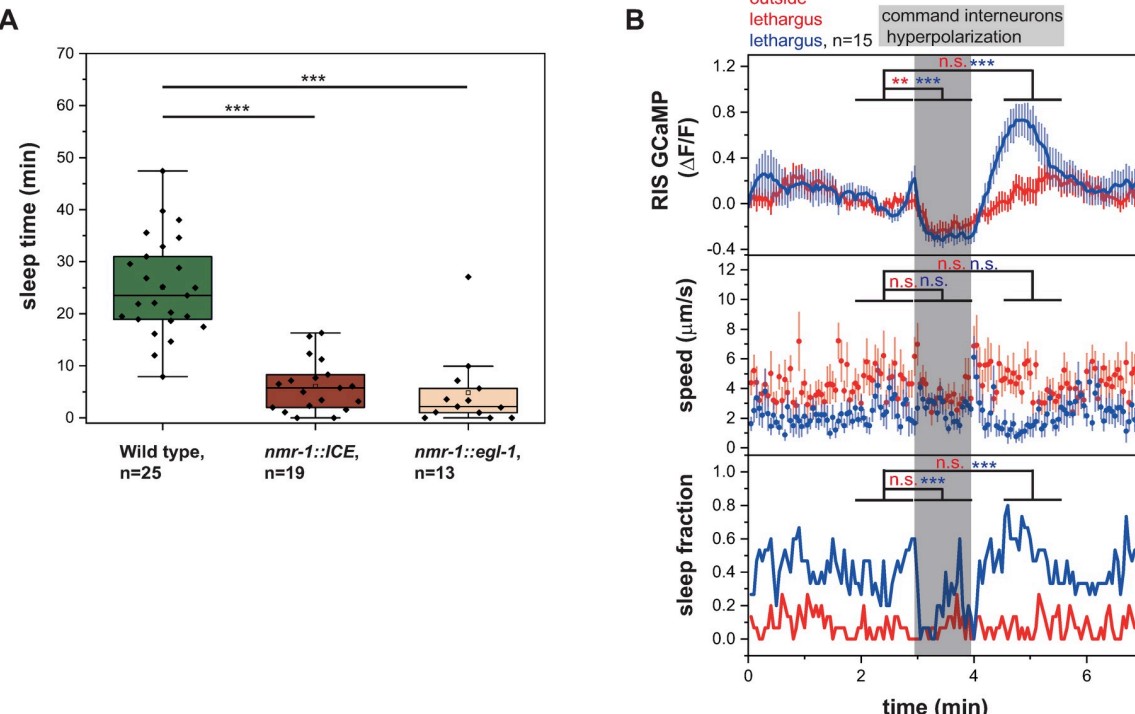

**Fig 5. The locomotion interneuron circuit controls RIS activation and sleep.** (A) Command interneurons are responsible for the majority of sleep. Command interneurons were genetically ablated by expressing ICE or *egl-1* under the *nmr-1* promoter. Command interneurons-ablated worms display a massive loss-of-sleep phenotype. ***$p < 0.001$, Welch test (S1 Data, Sheet 5A). (B) Hyperpolarization of command interneurons causes RIS inhibition and suppresses sleep. During lethargus, the hyperpolarization is followed by a strong post-stimulation activation of RIS. **$p < 0.01$, ***$p < 0.001$, Wilcoxon signed rank test for GCaMP and speed, Fisher's exact test for sleep fraction (S1 Data, Sheet 5B). ΔF/F, change of fluorescence over baseline; GCaMP, genetically encoded calcium indicator; ICE, Caspase-1/Interleukin-1 converting enzyme; n.s., not significant; RIS, Ring interneuron S class name.

Next, we wanted to test conditional loss of function of the command interneuron circuit on RIS activity. We expressed ArchT broadly in locomotion-controlling interneurons by using the *nmr-1* promoter. We then inhibited all command interneurons using green light and simultaneously imaged the activity of RIS. Inhibition of *nmr-1*-expressing neurons strongly inhibited RIS both outside and during lethargus. Interestingly, there was a strong post-stimulus activation of RIS, which was strongly increased only during lethargus. This activation peaked at approximately 170% of the RIS baseline. Sleep was inhibited by command interneuron inhibition, and worms reached mobility speeds similar to those outside of lethargus (Fig 5B). Mosaic analysis of an extrachromosomal array carrying the *nmr-1::ArchT* transgene revealed that RIS was partially inhibited when ArchT was expressed in head neurons but not in PVC and that the effect of inhibition was substantially stronger when ArchT was not only expressed in head neurons but also expressed in PVC (S9D and S9E Fig). This experiment showed that multiple interneurons act in concert to activate RIS and induce sleep. Among the *nmr-1*-expressing interneurons, only RIM and PVC are presynaptic to RIS [28,32]. However, additional reverse command interneurons could also contribute to RIS regulation through indirect mechanisms.

Because the command interneuron circuit is controlled by glutamatergic signaling [55,56] and because RIM activation of RIS requires glutamate (S3 Fig), we also analyzed the sleep behavior of *eat-4(ky5)* mutant larvae that have impaired glutamatergic neurotransmission. In

*eat-4(ky5)* mutant larvae, sleep-bout duration was significantly reduced, whereas sleep bouts occurred with normal frequency. This indicates that glutamate signaling might play a role in the maintenance but not in the initiation of sleep bouts (S10A–S10D Fig). Consistent with these findings, glutamate signaling also plays a role in the maintenance of NREM (non-Rapid Eye Movement) sleep in mice [13]. *nmr-1(ak4)* glutamate receptor mutant larvae only displayed slightly reduced RIS activation transients, which indicates that additional glutamate receptors are required for sleep induction (S10E–S10I Fig). Together, these mutant phenotypes support the view that excitatory neurotransmitter systems that are associated with locomotion are important for RIS activation.

## RIS inhibition causes homeostatic rebound activation

The design of the sleep circuit suggests an intimate mutual control mechanism of RIS and command interneurons that could allow homeostatic control of sleep. Arousing stimulation is known to inhibit sleep-active neurons and to increase subsequent sleep [22,23,25,27]. Consistent with these published data, we observed that the maximum RIS GCaMP intensity increased logistically with the length of the preceding motion bout during lethargus (S11A Fig). We thus hypothesized that stimulation inhibits RIS and leads to its subsequent depolarization, forming a homeostat that allows maintaining or reinstating sleep bouts. We tested this hypothesis by arousing the worms with a blue light stimulus (Fig 6A and 6B). During the stimulus, worms mobilized, and sleep was inhibited. In some of the trials, worms went back to sleep promptly after the stimulation and decreased their motion speed again within 3 min. Because worms did not remain mobile after the stimulation, we classified these trials as "nonmobilizing." In these nonmobilizing trials, RIS showed a post-stimulus activation, which was 34% stronger than the baseline activity. RIS activation correlated with a significantly increased fraction of sleep. In other trials during lethargus, the worms stayed mobile for at least 3 min after stimulation and did not go back to sleep. Because worms remained mobile after the stimulation, we classified these trials as "mobilizing." In these mobilizing trials, RIS stayed inhibited and was 16% less active than the baseline before stimulation (Fig 6A). To measure global neuronal activity during the blue-light stimulation experiment, we imaged worms that expressed pan-neuronal GCaMP [57]. Trials were again divided into mobilizing and nonmobilizing trials during lethargus depending on the mobilization status after the stimulus. Nonmobilizing trials showed a global neuronal inhibition that was 93% of the baseline activity (Fig 6B). These experiments show that noxious blue-light stimulation inhibits sleep and RIS and causes a reactivation of RIS when the system returns to sleep.

In normal sleep and in the sensory stimulation experiment, periods of inactivity of RIS were always followed by periods of RIS activation. This suggested that inhibition of RIS causes its subsequent reactivation. We tested this hypothesis by optogenetically hyperpolarizing RIS and following its activity using calcium imaging. We inhibited RIS directly for 60 s by expressing the light-driven proton pump ArchT specifically in this neuron and used green light illumination to activate ArchT. We followed RIS calcium activity using GCaMP during the experiment and quantified behavior. Optogenetic hyperpolarization of RIS led to a decrease in intracellular calcium and increased behavioral activity. Approximately 1 min after the end of the inhibition, RIS showed a rebound activation transient during which calcium activity levels increased strongly and rose well above baseline levels, concomitant with a decrease in behavioral activity. Overall brain activity measurements showed that behavioral activity and brain activity correlated throughout the experiment (Fig 6C). Rebound activation was observed neither following PVC nor following RIM inhibition (Fig 2C and S11B Fig), suggesting that rebound activation is specific to RIS and is not a general property of all neurons [58].

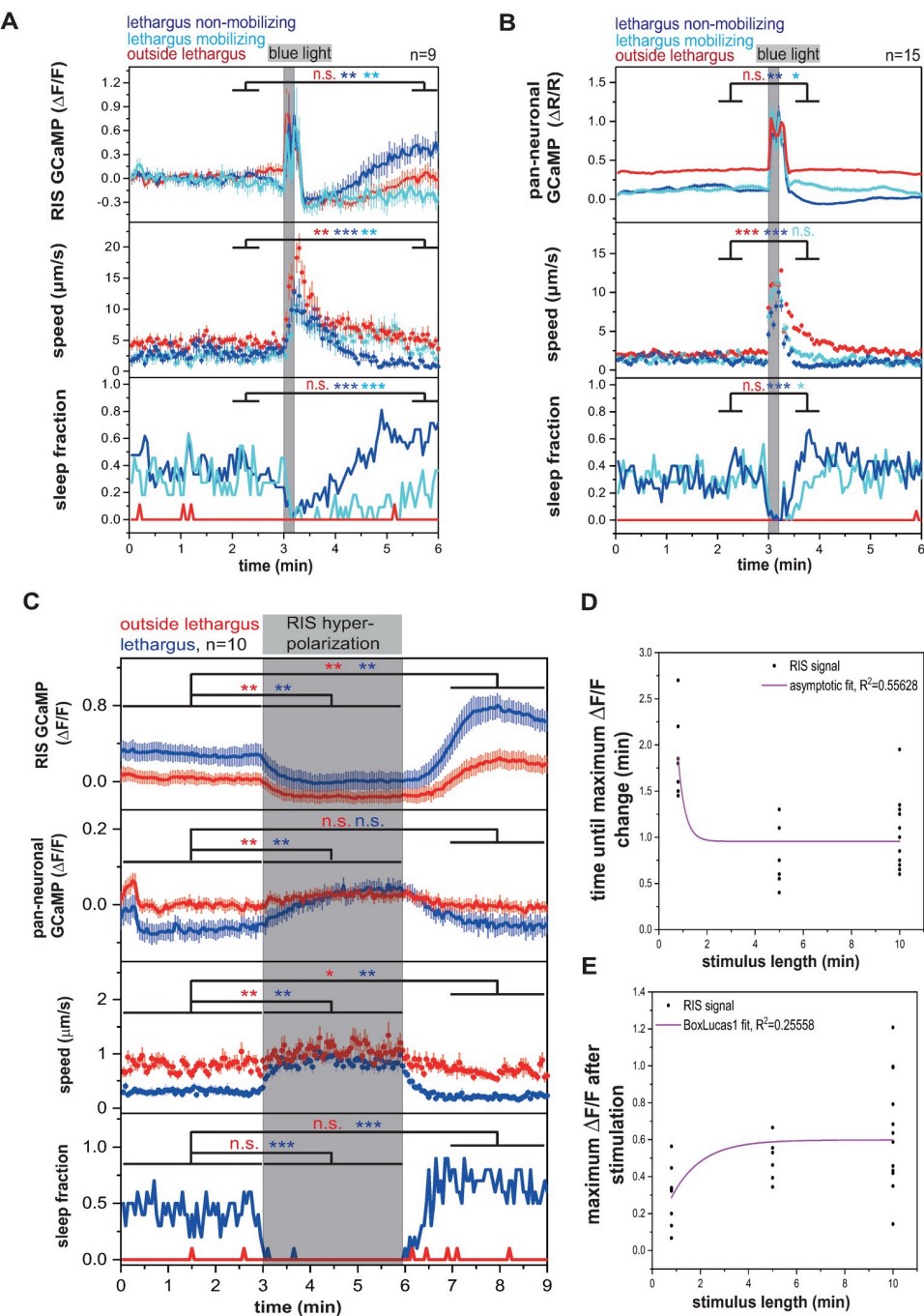

**Fig 6. RIS inhibition causes homeostatic rebound activation.** (A–B) A blue light stimulus leads to awakening and mobilization of *C. elegans*. Worms that go back to sleep after the stimulus show an activation rebound: pan-neuronal inhibition below baseline levels and RIS activation above baseline levels; "lethargus mobilizing" refers to animals that stayed awake and active during the post-stimulus time; "lethargus nonmobilizing" refers to animals that went back to sleep after the stimulation. $^*p < 0.05$, $^{**}p < 0.01$, $^{***}p < 0.001$, Wilcoxon signed rank test for GCaMP and speed, Fisher's exact test for sleep fraction (S1 Data, Sheet 6A and 6B). (C) RIS shows rebound activation following hyperpolarization. Behavioral and brain activity measurements correlate throughout the whole experiment. $^*p < 0.05$, $^{**}p < 0.01$, $^{***}p < 0.001$, Wilcoxon signed rank test for GCaMP and speed, Fisher's exact test for sleep fraction (S1 Data, Sheet 6C). (D–E) Dose-response curve of optogenetic RIS hyperpolarization with different stimulus lengths. RIS activation rebound transients saturate with increasing length of inhibition. Worms not showing a rebound activation transient after RIS optogenetic hyperpolarization were excluded from the analysis. Numbers of worms not responding were as follows: (1) In experiments in which RIS was optogenetically inhibited for 48 s, all worms showed an RIS

rebound activation transient. (2) In experiments in which RIS was optogenetically inhibited for 5 min, 1 out of 7 worms did not show a RIS rebound activation transient. (3) In experiments in which RIS was optogenetically inhibited for 10 min, 1 out of 13 worms did not show an RIS rebound activation transient. Curve in D was fitted as an asymptotic function, and curve in E was fitted as a BoxLucas1 function (S1 Data, Sheet 6D, E). ΔF/F, fluorescence change over baseline; GCaMP, genetically encoded calcium sensor; n.s., not significant; R, fluorescence of GCaMP divided by fluorescence of mKate2; RIS, Ring Interneuron S class name.

Strikingly, while the rebound transient was also measurable outside of lethargus, the strength of the RIS rebound depolarization was 3-fold stronger during lethargus than before lethargus, indicating that the propensity for RIS rebound activation is strongly increased during lethargus.

To test whether rebound activation of RIS mediates acute or chronic homeostasis, we tested whether the strength of the rebound activation is a function of length of prior inhibition. For this experiment, we increased the length of the RIS inhibition and quantified the time it took after the end of the stimulation until the rebound transient started as well as the peak maximum of the rebound. After inhibiting RIS for 5 min, the rebound initiated immediately after the end of the stimulation and the maximum that was reached exceeded that observed after about 1 min of RIS stimulation. Inhibiting RIS for 10 min did not further increase the occurrence or strength of the rebound transient. These results show that RIS activation rebound transients rapidly saturate with increasing length of inhibition (Fig 6D and 6E and S11C–S11E Fig). Thus, RIS shows a rebound activation following inhibition. The rebound activation presents the translation of RIS inhibition into subsequently increased RIS activity and thus sleep induction. Rebound activation of RIS does not seem to constitute a chronic integrator of wake time but presents an acute homeostatic regulatory phenomenon to induce or reinstate sleep bouts.

Rebound activation of RIS could present a cell-intrinsic property or could be generated by a neural circuit. To discriminate between these hypotheses, we measured rebound activation in *unc-13(s69)* mutant animals in which synaptic signaling is globally impaired [59], or in worms that express tetanus toxin [60] specifically in RIS to abrogate synaptic transmission specifically in this neuron. Rebound activation of RIS was abolished in RIS::tetanus toxin (S11F–S11G Fig) as well as *unc-13(s69)* worms (S11F and S11H Fig). These results indicate that rebound activation of RIS is a property of the neuronal network.

In analogy to the activation rebound seen after optogenetic RIS inhibition, optogenetic RIS activation might cause a negative rebound, i.e., an inhibition of RIS inhibition below baseline levels following its optogenetic activation. Indeed, we observed such an effect. Interestingly, the negative rebound was 3-fold stronger during lethargus compared to outside of lethargus (S11I Fig). However, such a negative rebound was also present in other neurons such as PVC (S7A Fig), making it difficult to judge whether this effect is part of a specific sleep homeostatic system or rather a general response of neurons to strong depolarization [58]. In summary, RIS activity is homeostatically regulated, with its inhibition causing its reactivation. This rebound activation is strongly increased during lethargus and likely is required for inducing or reinstating sleep.

## Modest dampening of brain arousal occurs upstream of RIS

Our results demonstrate that the command interneuron circuit, including PVC, plays a major role in activating RIS involving self-enforcing positive feedback, resulting in strong RIS activation and thus sleep induction. RIS calcium transients are small during development outside of lethargus, whereas transients are high during lethargus. What determines that RIS calcium transients are limited outside of lethargus but promoted during lethargus? As an important

principle of command interneuron control, forward and reverse command interneurons inhibit each other to allow discrete forward and reverse locomotion states. The AVA/AVD/AVE/RIM interneurons initiate reverse locomotion by activating premotor interneurons while inhibiting the forward command circuit including AVB/PVC. By contrast, during forward movement, reverse command interneurons are inhibited [49,56].

Small changes in arousal and activity of the command interneurons can change the equilibrium of forward and reverse command interneurons [55]. Hyperactive mutants suppress sleep across species, including *C. elegans* [61–68]. Many arousal cues trigger backwards escape movements and inhibit RIS [25,27,69]. Thus, previous studies on the command interneuron circuit together with our results suggest that arousal inhibits RIS through inhibiting PVC. This model of RIS activation would predict that there are changes during lethargus that are upstream of RIS activity that change the properties of the command circuit, leading to increased PVC and thus RIS activation.

We reasoned that it should be possible to measure these changes that occur in command interneuron activity upstream of RIS by characterizing neural activity and behavior in *aptf-1(−)* mutant worms. We quantified behavior and command interneuron calcium levels across lethargus in *aptf-1(−)* mutant worms. Wild-type animals showed successive sleep bouts and a 72% reduction in locomotion speed during lethargus. By contrast, *aptf-1(−)* mutant animals almost never showed quiescence bouts (Fig 3I), but nevertheless, locomotion speed was decreased by 20% during the lethargus phase (Fig 7). Consistent with the behavioral activity reduction, there was a significant reduction of command interneuron activity during lethargus also in *aptf-1(−)* mutant animals (Fig 7 and S12 Fig). To further characterize the neuronal changes upstream of RIS-mediated sleep induction, we imaged the activity of RIM during lethargus in *aptf-1(−)* mutants. In wild-type animals, RIM regularly showed activation transients before lethargus but did not show many transients during lethargus. RIM showed not only a change in transient frequency across the lethargus cycle but also a reduction in baseline calcium activity. In *aptf-1(−)* mutant worms, RIM continued showing calcium transients during lethargus, indicating that RIS inhibits calcium transients in RIM during sleep bouts. However, reduction of baseline calcium activity was preserved in *aptf-1(−)*, indicating that RIM activity is dampened during lethargus independently of RIS at the level of baseline calcium activity. Together, these experiments indicate that a dampening of behavioral and neural baseline activity that is independent of RIS occurs during lethargus. This neuronal baseline and behavioral dampening itself appears not to be sufficient to constitute normal sleep bouts but could hypothetically lead to an activity change and decreased mutual inhibition in command interneurons, thus promoting sleep induction [55,70].

## An arousing stimulus inhibits RIS through RIM

Arousal plays a major role in inhibiting sleep, but the circuits that mediate the effect of arousing stimuli on RIS inhibition are not well understood. We therefore studied the circuit by which stimulation of a nociceptor, the ASH neurons, leads to a reverse escape response and inhibition of RIS [71]. We optogenetically stimulated ASH using ReaChR and green light and followed RIS and RIM activities. ASH activation led to a strong activation of the RIM neuron and triggered a backwards response as previously described [35,71]. Simultaneously, RIS was inhibited (Fig 8A). RIM can inhibit PVC through reverse interneurons that it synchronizes [49,72]. Furthermore, strong RIM activation can inhibit RIS more directly. To test whether ASH indirectly inhibits RIS through RIM, we ablated RIM genetically by expression of *egl-1* from the *tdc-1* promoter [35,38] and repeated the optogenetic stimulation of ASH. In RIM-ablated L4 animals, activation of ASH caused the opposite effect on RIS activity. Instead of

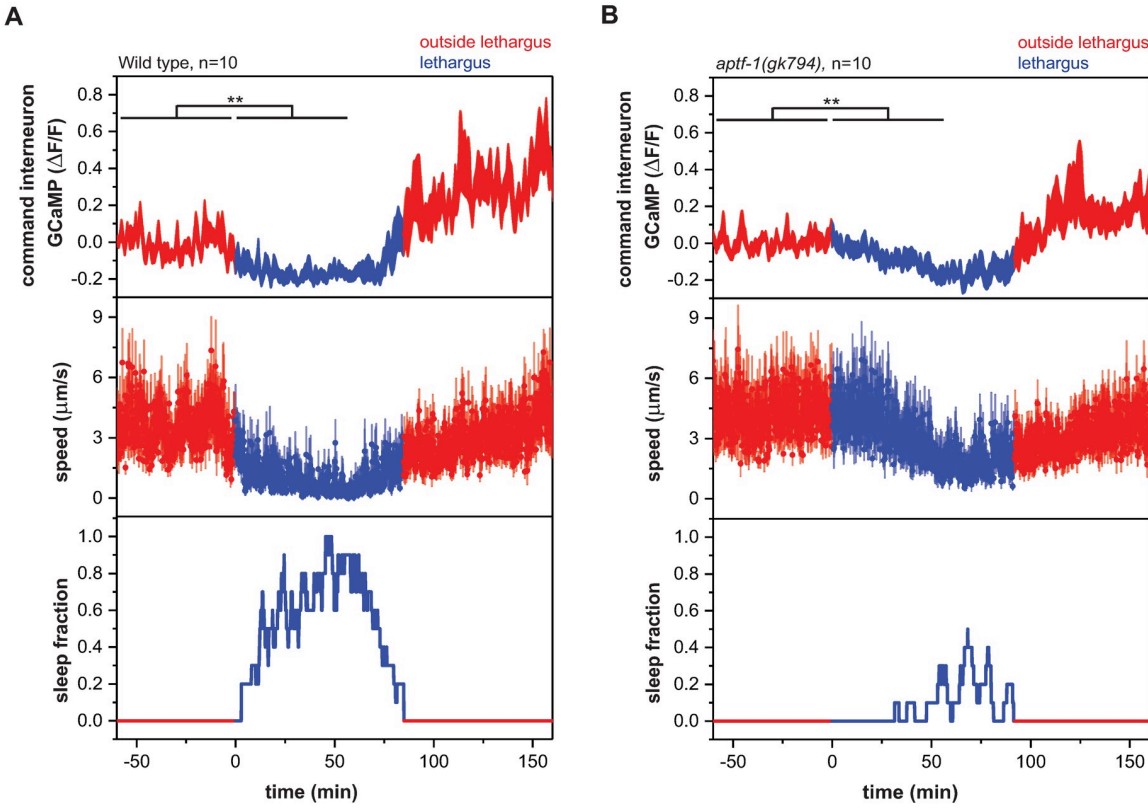

**Fig 7. The dampening of neural and behavioral baseline activity levels during lethargus is independent of RIS function.** Reduction of command interneuron activity levels during lethargus occurs in wild-type worms and *aptf-1(gk794)* mutants. In the wild-type condition, activity levels are reduced to −0.16 ± 0.02. In the mutant condition, activity levels are reduced −0.08 ± 0.02. ** $p < 0.01$, Wilcoxon signed rank test (S1 Data, Sheet 7A and 7B). ΔF/F, fluorescence change over baseline; GCaMP, genetically encoded calcium indicator; RIS, Ring Interneuron S.

inhibiting RIS, ASH activated RIS, while it still increased behavioral activity (Fig 8B). Consistent with our calcium imaging data, ASH stimulation after RIM ablation predominantly caused a forward locomotion response (Fig 8C). There are 2 ways ASH might inhibit RIS through RIM. One possibility is that arousal strongly activates reverse interneurons, thus inhibiting forward PVC neurons and RIS during stimulation. Consistent with this idea, gentle tail touch increased RIS activity more strongly when RIM was ablated (S13 Fig). Another option is that RIM inhibits RIS directly through tyramine and FLP-18. Both circuits might play together (Fig 8D). These results delineate a circuit model for how sensory stimulation can control RIS activation.

## Discussion

### A wake-active circuit that controls locomotion also controls sleep

Optogenetic activation and inhibition showed how the activity of presynaptic neurons affects RIS depolarization during developmental sleep. Several presynaptic neurons can activate RIS. RIM appears to be a potent direct inhibitor when activated strongly but can also act as an activator of RIS. Loss-of-function experiments showed that the command circuit controls activation of RIS, with PVC presenting a key activator of RIS. PVC has long been known to mediate the forward escape response by transmitting information from posterior sensory neurons to

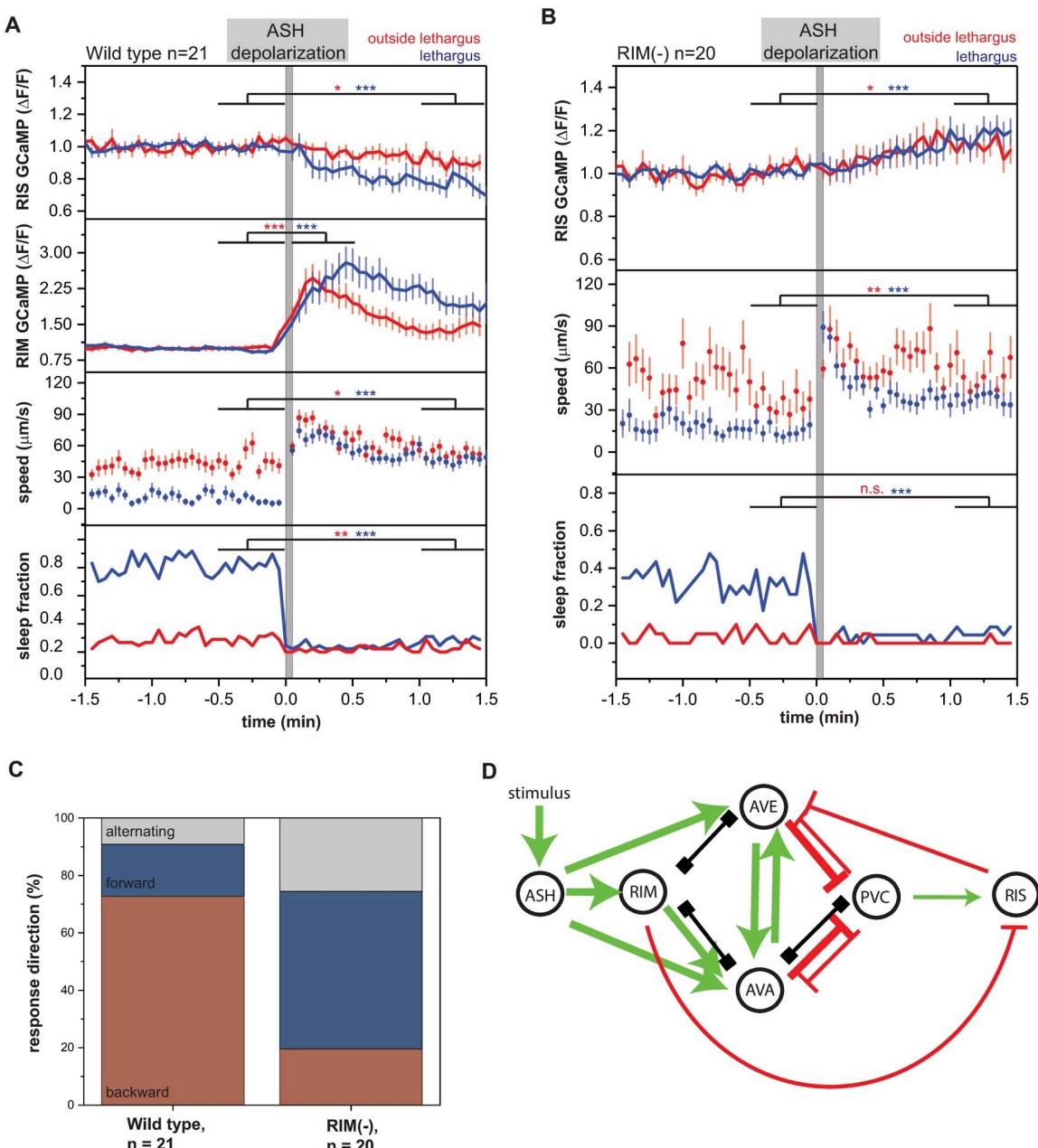

**Fig 8. Arousing stimulation inhibits RIS and sleep through RIM.** (A) ASH depolarization in wild-type worms leads to RIS inhibition and RIM activation, sleep suppression, and mobilization. $^*p < 0.05$, $^{**}p < 0.01$, $^{***}p < 0.001$, Wilcoxon signed rank test for GCaMP and speed, Fisher's exact test for sleep fraction (S1 Data, Sheet 8A, B). (B) ASH depolarization in RIM-ablated worms leads to weaker sleep suppression, mobilization, and RIS activation. $^*p < 0.05$, $^{**}p < 0.01$, $^{***}p < 0.001$, Wilcoxon signed rank test for GCaMP and speed, Fisher's exact test for sleep fraction (S1 Data, Sheet 8A, B). (C) The response direction following ASH activation in wild-type worms is predominantly reverse, while in RIM-ablated worms it is predominantly forward. $^{***}p < 0.001$, Fisher's exact test (S1 Data, Sheet 8C). (D) A circuit model for RIS regulation through arousal by ASH. Activating synaptic input is shown as green arrows, inhibitory synaptic input is shown as red arrows, and gap junctions are indicated as black connections. RIM could serve as a synchronizer of AVE and AVA to regulate PVC and therefore RIS inhibition. Additionally, RIM could inhibit RIS directly. ΔF/F, fluorescence change over baseline; GCaMP, genetically encoded calcium indicator; n.s., not significant.

activate AVB premotor neurons to trigger forward locomotion [46,49,50]. Consistent with promoting the forward escape response, optogenetic activation of PVC leads to an increase in forward movement [50,73] (Fig 2D). Reverse movement, in turn, is mediated by AVA, AVE, and AVD command premotor interneurons, which activate reverse motor neurons. Forward PVC and reverse AVA/AVE/AVD command interneurons are presynaptic to and mutually inhibit each other, which ensures discrete forward and reverse locomotion states analogous to a flip-flop switch [49,55,74].

Our finding that PVC and RIM neurons present key activators of RIS that act in concert suggests a model for how RIS is controlled; it also provides a potential mechanism for linking sleep induction to decreasing arousal and for homeostatically maintaining a series of sleep bouts. According to this model, during conditions of high arousal, such as during development outside of lethargus, larvae are constantly awake. The command interneuron circuit cycles between forward and reverse states, leading to the activation of forward or reverse motor programs, respectively [49,74,75]. PVC activation has been associated with the activity of forward states, and RIM has mostly been associated with the activity of reverse states. Because neither activation of only the PVC nor of the RIM neurons appears to be sufficient for sleep induction, RIS should not be activated sufficiently to induce sleep during either forward or reverse states. At the transition between forward and reverse states, locomotion pauses can occur. It has been shown that, in adult worms, RIS shows activation transients in the nerve ring during locomotion pauses. These calcium transients appear to be much smaller compared with activation transients during sleep bouts that extend to the cell soma. Locomotion pausing is reduced after RIS ablation, suggesting that weak RIS activation promotes pausing [76].

Lethargus induces a modest dampening of neuronal baseline activity that is independent of RIS and that includes the RIM neurons. The RIM neurons become less excitable, which should reduce their inhibitory effects on RIS and instead favor their activating effects. PVC becomes resistant to inhibition and more potent in its capacity to activate RIS. We hypothesize that these shifts in the properties of the interneurons of the locomotion circuit favor the activation of the RIS neuron. RIS activation appears to require concerted activation from PVC and RIM neurons (a process that is perhaps aided by other locomotion interneurons). Both PVC and RIM appear to depolarize prior to RIS activation, and both types of neurons contribute to RIS depolarization. This suggests that RIS might be activated when both PVC and RIM exert activating effects. Such an overlapping activating effect of PVC and RIM on RIS would most likely occur at the transition from forward to reverse locomotion states, where there could be an overlap of both forward and reverse neuronal activities. This would suggest that both locomotion stop and sleep bouts might be induced by locomotion control interneurons at the transition between forward and reverse locomotion states. The difference between locomotion stop and a sleep bout would be that, in the former, RIS would only be modestly activated, whereas in the latter, RIS would be strongly activated (Fig 9). Consistent with this model, sleep bouts are typically induced at the end of long forward movements, whereas the exit from the sleep bout—e.g., caused by a noxious stimulus—is often through a reverse movement [70,75,77]. Arousal promotes reverse command interneuron activity and strong RIM activation that can inhibit RIS. Locomotion control and periods of behavioral activity and rest are already present in animals that do not have a nervous system. It has therefore been hypothesized that sleep and sleep-active neurons evolved from systems controlling locomotion activity and rest [8]. The finding that a sleep-active neuron can also act as a locomotion pause neuron [76]—and the discovery presented here that the locomotion circuit controls the depolarization of a sleep-active neuron—suggests that sleep-controlling circuits might have evolved from locomotion-controlling circuits and therefore that locomotion quiescence and sleep could be regarded as homologous behaviors.

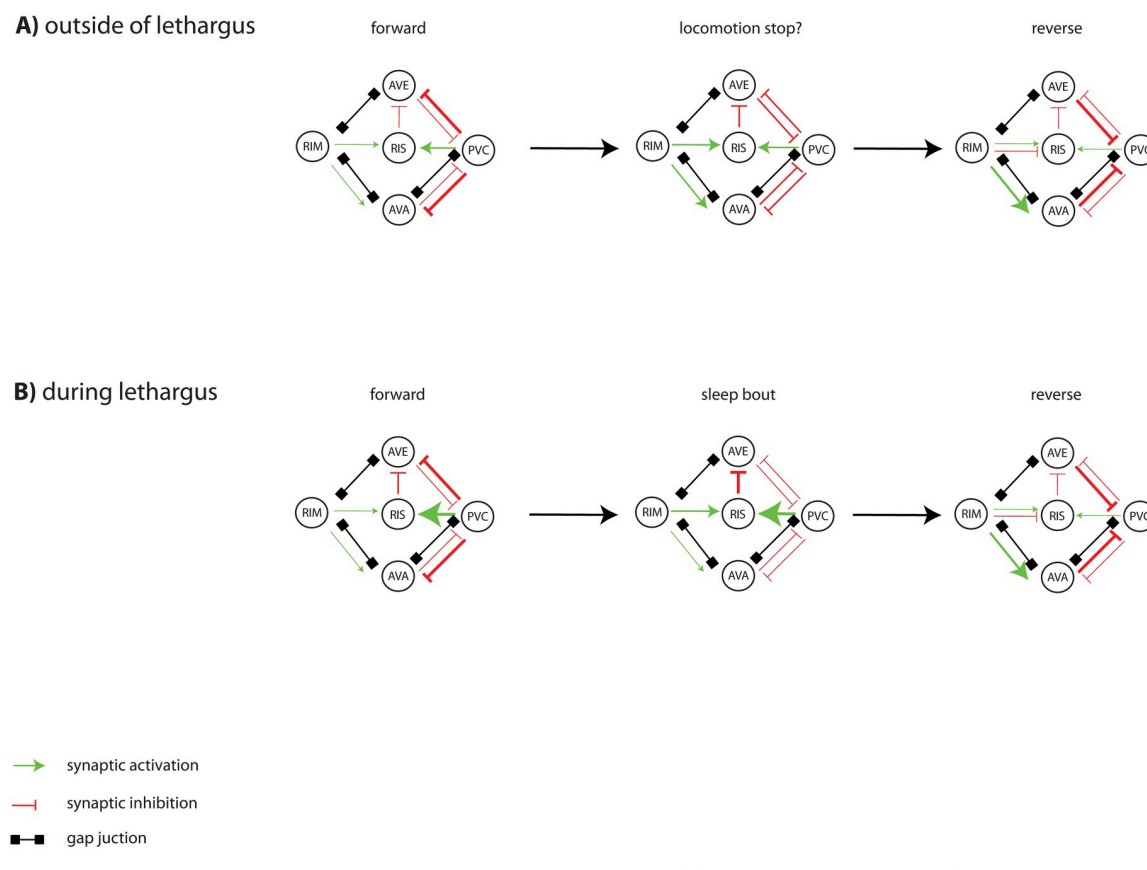

**Fig 9. A circuit model for RIS activation through locomotion interneurons.** (A) Activating synaptic input is shown as green arrows, inhibitory synaptic input is shown as red arrows, and gap junctions are indicated as black connections. Outside of lethargus, the nervous system cycles between forward and reverse states. RIS is not activated sufficiently to cause a sleep bout, neither during the forward state during which PVC is active nor during the reversal state during which RIM is active. The locomotion circuit activates RIS briefly to cause a locomotion pause at the transition from forward to reverse movement. Speculatively, the circuit that controls RIS during sleep also controls RIS during locomotion pauses. (B) During lethargus motion bouts, the nervous system still cycles between forward and reverse states. Baseline activity and excitability in RIM are reduced, and PVC becomes resistant to inhibition and more potent to activate RIS. These changes in locomotor interneurons shift the balance to favor strong RIS activation and induction of a sleep bout, a process that may involve simultaneous activation from multiple neurons, including RIM and PVC. Such an overlap activation of RIS by otherwise mutually exclusive neurons could occur at the transition from forward to reverse locomotion states. Perhaps, RIS activation and sleep could occur similarly at the transition from reverse to forward locomotion states.

Our model suggests that the sleep switch is tripartite and includes not only wake-active wake-promoting neurons and inhibitory sleep-active sleep-promoting neurons but also wake-active sleep-promoting neurons as mediators of switch flipping. This sleep switch acts as an amplifier that can translate a modest reduction of arousal into a massive shutdown of behavioral activity during sleep. Dampening of neural activity and altered properties of wake-active sleep-promoting locomotion neurons independently of sleep-active neurons could be interpreted as a neural equivalent of sleepiness that leads to an increased propensity to activate sleep-active neurons and to induce sleep bouts.

Mutations that increase arousal and suppress sleep increase the activity of reversal neurons, whereas conditions that decrease arousal decrease the activity of the reversal neurons and therefore increase the amount of sleep [67,68,74]. Also, the ablation of reverse command interneurons such as AVE reduces reversals and leads to ectopic quiescence, as well as increases sleep [46,52] (and this study). According to our model, increasing arousal should increase the activity of RIM and other reverse command interneurons and thus should inhibit RIS.

Conversely, reducing arousal could promote weaker RIM activation and PVC activation that should shift the equilibrium to stronger RIS activation.

What causes the termination of sleep bouts? The RIS neuron might not be able to sustain prolonged activity, leading to the spontaneous cessation of a sleep bout. The RIS activation transient and thus sleep bout can be blunted prematurely by a sensory or optogenetic arousing stimulus [25,27,70,78]. Arousing stimulation, for instance, by activating the nociceptive sensory neurons, triggers a reverse escape response through backwards command and RIM interneurons [35,72,75,79]. Strong optogenetic RIM depolarization inhibits RIS, and stimulation of the nociceptive ASH neurons causes inhibition of RIS that depends on RIM, suggesting that a main physiological role of strong RIM activation is to inhibit sleep upon arousing stimulation, perhaps by synchronizing the reverse interneurons [72]. RIM activation can inhibit sleep also in response to acute food deprivation [80,81]. Thus, RIM might present not only an activator of RIS but also an arousal module that can be activated upon sensing various external conditions that signal the need to suppress sleep.

RIS inactivation leads to disinhibition of arousal and brain activity, starting anew the cycle of locomotion interneuron activity and locomotion behavior. Depending on the arousal levels, the locomotion circuit causes RIS reactivation and thus a return to sleep either immediately or after a delay. The timing of the rebound activation can be controlled by the level of arousal—with strong arousal leading to longer wake periods before the return to sleep—whereas milder stimulations cause an immediate return to sleep [23]. Consistent with this circuit model of recurrent RIS activation, RIS activity oscillates, resulting in the typical pattern of sleep bouts that are interrupted by activity bouts [22]. This circuit design allows homeostatic sleep maintenance of a series of consecutive sleep bouts with sensory stimulation restarting the cycle of RIS activation, thus prompting an acutely increased RIS activation causing the return to sleep (Fig 6A) [23,70]. Our model predicts that RIS calcium transient strength is a function of prior behavioral activity. Consistent with this view, RIS calcium transients are stronger at the beginning and end of lethargus, when motion bouts are high, but are less pronounced in the middle of lethargus, when motion bouts are less pronounced (Fig 1A) [22,23]. Thus, the tripartite flip-flop circuit design allows an adaptation of RIS activity to the strength required to induce sleep bouts at a given behavioral activity level.

Here, we have identified a circuit controlling sleep-active neuron depolarization in *C. elegans*. This work built on the neural connectome and was facilitated by the small size and invariance of the nervous system as well as the transparency of the organism. While the *C. elegans* sleep circuit clearly is built from fewer cells than the mammalian sleep circuit [8,82,83], there are many conceptual similarities. For instance, in both *C. elegans* and humans, sleep is controlled by inhibitory sleep-active sleep-promoting neurons that depolarize at sleep onset to actively induce sleep by inhibiting wake circuits. A main difference is that humans have many brain centers each consisting of thousands of sleep-active neurons [12]. The single RIS neuron is the major inhibitory sleep-active neuron required for sleep induction in *C. elegans* [25]. Work in mammals revealed the general principles of wake-active wake-promoting neurons and sleep-active sleep-promoting neurons as well as their mutual inhibition. While this information explains the flip-flop nature of sleep and wake states, there is no satisfactory understanding of what flips the sleep switch, i.e., how wakefulness is detected when the system is set to sleep, prompting the activation of inhibitory sleep-active neurons [6]. Our model for the operation of the *C. elegans* sleep circuit indicates that flipping of the sleep switch can be understood if wake-active sleep-promoting neurons are added to the switch model. In this tripartite flip-flop sleep switch model, the sleep-active sleep-promoting center is activated by wake-active neurons. This activation should, however, only occur when the system is set to sleep, a state that could present a neural correlate of sleepiness.

Sleep is reversible by stimulation, and hyperarousal is the major cause for insomnia in humans [3,84,85]. Homeostatic sleep maintenance is an essential feature of sleep and is found from worms to humans [19–21,23]. R2 ring neurons in *Drosophila* present an integrator of wake time, causing subsequently increased depolarization of dFB sleep-inducing neurons, thus forming a chronic sleep homeostat [86,87]. In vertebrates, serotonergic raphe neurons are active during wakefulness and can reduce behavioral activity and increase sleep pressure [88]. Our model of a tripartite flip-flop circuit suggests that wake-active sleep-promoting neurons are an essential part of an acute sleep homeostat that translates acute brain activity into increased sleep neuron activity when the system is set to sleep. Wake-active sleep-promoting neurons measure systemic activity, i.e., they become active together with a global brain activity increase and can then activate inhibitory sleep-active neurons. Thus, the interplay of sleep-active sleep-promoting and wake-active sleep-promoting neurons form an oscillator that periodically sends out sleep-inducing pulses. Macroscopically, sleep in mammals exists as cortical oscillations of global down states, known as slow waves [89]. Micro-arousals trigger cortical up states that are followed by cortical down states, known as k-complexes [19–21]. Both slow-wave activity as well as k-complexes could be hypothetically generated by wake-active sleep-promoting neurons.

## Materials and methods

### Worm maintenance and strains

*C. elegans* worms were grown on Nematode Growth Medium (NGM) plates seeded with *Escherichia coli* OP50 and were kept at 15 ˚C to 25 ˚C [90]. Crossed strains were genotyped through Duplex PCR genotyping of single worms [91]. The primer sequences that were used for Duplex PCR can be found subsequently. To confirm the presence of transgenes after crossings, fluorescent markers were used. All strains and primers that were used in this study can be found in S1 Text and S1 Table.

### Strain generation

DNA constructs were cloned with the 3-fragments Gateway System (Invitrogen, Carlsbad, CA) into pCG150 to generate new strains [92]. The *ArchT*, the *ReaChR*, and the *egl-1* genes were expression optimized for *C. elegans* [93]. The *tdc-1*::*egl-1* transgene specifically expresses the apoptosis-inducing protein EGL-1 in RIM and RIC. Therefore, RIM and RIC are genetically ablated in worms carrying this transgene. The ablation is probably incomplete in L1 worms. The *nmr-1*::*egl-1* transgene leads to the expression of *egl-1* in all command interneurons causing their genetic ablation. Similar to the *tdc-1*::*egl-1* transgene, ablation might be incomplete in L1 worms. In both lines, *egl-1* was co-expressed with *mKate2*, which was used to verify the genetic ablations. Transgenic strains were generated by microparticle bombardment or by microinjection. For microparticle bombardment, *unc-119(ed3)* was used. The rescue of the *unc* phenotype was therefore used as a selection marker [94,95]. The transgenes were back-crossed twice against N2 wild-type worms to remove the *unc-119(ed3)* background. Extrachromosomal arrays were generated by DNA microinjection. DNA was injected in wild-type, mutant, or transgenic worms. For injection, DNA was prepared as follows: construct 30–100 ng/μl, co-injection marker 5–50 ng/μl, and pCG150 up to a concentration of 100 ng/μl if required. Positive transformants were selected according to the presence of co-injection markers. A table of all plasmids and a list of all constructs that were generated for this study can be found in S2 Table and S2 Text.

## Generation of gene modifications using CRISPR

The following allele was designed by us in silico and was generated by SunyBiotech. Correctness of the alleles was verified by using Sanger sequencing.

PHX816: *flp-11(syb816 [SL2::mKate2::linker(GSGSG)::tetanustoxin_LC]) X*

The coding sequences of tetanus toxin light chain and mKate2 were codon optimized and intronized as described previously and were synthesized [93]. The final sequence can be found in S3 Text.

## Imaging

**Cameras and software.**   All imaging experiments were conducted using either an iXon EMCCD (512 × 512 pixels) (Andor Technology Ltd., Belfast, UK), an iXon Ultra EMCCD (1,024 × 1,024 pixels) (Andor Technology Ltd.), a Photometrics Prime 95B back-illuminated sCMOS camera (1,200 × 1,200 pixels) (Nikon, Tokyo, Japan), or a Nikon DS Qi2 (4,908 × 3,264 pixels) (Nikon). For the iXon cameras, the EM Gain was set between 100 and 200. The exposure times used were between 5 and 30 ms. Andor IQ 2 and 3 (Andor Technology Ltd.) and NIS Elements 5 (Nikon) were used for image acquisition.

**Illumination systems.**   A standard 100-W halogen lamp together with an infrared filter (Semrock Brightline HC 785/62) (Idex Health and Science, New York) were used for differential interference contrast (DIC) microscopy or bright-field imaging. For calcium imaging and blue light stimulation, an LED illumination (CoolLED) with a 490-nm LED and standard GFP filter set (EGFP, Chroma) were used. Optogenetic stimulations and RFP imaging were performed with an LED illumination (CoolLED) with a 585-nm LED and standard TexasRed filter set (Chroma Technology Corp., Bellows Falls, VT).

## Agarose microchamber imaging

Long-term imaging experiments were conducted in agarose microchambers as previously described [96,97]. To summarize, a PDMS mold was used to cast box-shaped indentations in a hydrogel, which consisted of 3% or 5% agarose dissolved in S-Basal [98]. Two different sizes were used. We imaged L1 larvae in 190 × 190 × 15 μm microchambers, and L4 larvae were imaged in 370 × 370 × 25 μm microchambers. Depending on the developmental state of the worm that was imaged, either pretzel-stage eggs or L3 larvae were picked into the chambers with OP50 bacteria. Before imaging, worms were kept at either 20 ˚C or 25 ˚C.

For time-lapse calcium imaging experiments, L1 worms were filmed every 5 s (Figs 1A, 3H–3J, 4B–4D, 5A, S8C–S8F and S11A Figs), every 8 s (Fig 7, S9A, S9C and S10 Figs), or every 10 s (Figs 3F and 4A) with DIC or bright-field imaging and widefield fluorescence. The DIC and bright-field light source was left on continuously, filtered through an infrared filter, and was blocked by a shutter during fluorescence image acquisition. LED illumination was triggered by the EMCCD camera using the TTL exposure output of the camera. An objective with 20× magnification, an LED with 480 nm (light intensity was between 0.15 and 2 mW/mm$^2$), and EM gain of 100–200 was used. With the 20× objective and a 0.7 lens, 4 worms could be imaged simultaneously in one field. One to four fields could be filmed in parallel in one experiment. These image sequences gave measurable neuronal calcium transients and clear DIC or bright-field images to identify pumping or nonpumping phases.

During the continuous experiments in Figs 3F, 4B–4D and 5A, only DIC or bright-field images were taken.

## AVA inhibition experiment

NGM plates were prepared with histamine (HA; Sigma Aldrich, St. Louis, MO, 10 mM) as previously described [54]. Young adult worms expressing a HA chloride channel in AVA and control worms were picked onto NGM HA plates the night before the experiments. The next morning, eggs together with *E. coli* bacteria from the NGM HA plates were picked into microfluidic chambers and DIC imaged as previously described [96,97].

## Optogenetic experiments

Optogenetic experiments were either conducted in agarose microchambers as described previously, or the worms were immobilized. For immobilization experiments, the agarose was solved in S-Basal. We used the following 3 methods of immobilization for optogenetic experiments:

1.  Immobilization on a 3% agarose pad with 25 mM Levamisole (Sigma Aldrich) (S6 Fig)

2.  Immobilization on a 10% agarose pad with 0.1 μm Polybead microspheres (Polysciences, Warrington, PA) [99] (Fig 1B/ SDQL, Fig 1C/ SDQL, Fig 3B–3D, S1A Fig/SDQL, S5A Fig/ SDQL, S4 and S8A Figs)

3.  Immobilization on a 10% agarose pad with 0.1 μm Polybead microspheres [99] and 25 mM Levamisole (Fig 1B/PVC, Fig 1C/PVC, Figs 2A, 2C, 3B–3D, 4F–4G, S1A Fig/PVC, S5A Fig/ PVC, S7A, S7C, S7D, S8B and S8H Figs)

Worms were imaged within 30 min of immobilization. A 100× oil objective was used for illumination and imaging in most experiments. For images in S6A and S6B Fig, a 1.5 lens was added (S6A and S6B Fig). The imaging in Fig 2A was done using a 40× objective.

ReaChR for neuronal depolarization or ArchT for hyperpolarization was utilized. For optogenetic stimulation, a 585-nm LED and a standard TexasRed filter set were used.

For optogenetic experiments with L1 larvae, either L4-stage worms or young adult worms were prepicked onto NGM plates with all-trans-retinal (ATR; Sigma Aldrich) and grown at 20 ˚C or 25 ˚C. During the 2 d after exposure to ATR, pretzel-stage eggs or L1 worms were taken from this plate for optogenetic experiments. For optogenetic experiments with L4 larvae, an agar chunk containing a mixed population of growing worms was added to NGM plates containing ATR. Worms for optogenetic experiments were taken from this plate within the next 2 d.

Calcium imaging was conducted with an interval of 3 s and with an exposure time of 5–200 ms. A standard optogenetic protocol included calcium imaging during a baseline. This was followed by a stimulation time, in which the worms were optogenetically stimulated. The 585-nm light exposure was continuous except for brief interruptions during the time calcium imaging was conducted. After the optogenetic stimulation, calcium images were acquired during a recovery period.

In mobile worms, this standard protocol was preceded by 20 DIC frames that were taken every 500 ms to determine whether the worm was pumping. The overall protocol was repeated every 15 to 30 min. L1 mobile worms were imaged with a 20× objective and a 0.7 lens. Mobile L4 worms were imaged with either a 10× objective (Fig 8A–8C) or a 20× objective (Fig 1B/ CEP, Fig 1C/URY, S1A Fig/CEP, S5A Fig/URY and S3 Fig). Fixed worms were usually imaged between 1 and 4 trials. A delay preceded the standard protocol to allow the worm to recover from immobilization and between trials. To specifically manipulate PVC and SDQL in Fig 1B/ PVC, Fig 1B/SDQL, Fig 1C/PVC, Fig 1C/SDQL, Fig 2D, S1A Fig/PVC, S1A Fig/SDQL, S4 Fig, S5A Fig/PVC, S5A Fig/SDQL, and S7C and S7D Fig, the stimulating illumination was

restricted to the neuronal areas. This was achieved by reducing the size of the field aperture of the fluorescence illumination. To activate a specific neuron, it was moved into the illuminated area by using an automated stage. To image RIS, this neuron was moved into the illuminated area by the automated stage, while the optogenetic light stimulus was switched off and imaging light was switched on. The details for optogenetic experiments can be found in S3 Table.

### Behavioral imaging during PVC activation

Worms were prepared on retinal plates and picked into microchambers as described previously. A 20× objective was utilized for imaging. The entire chamber was imaged through bright-field imaging. For tail-specific illumination, the LED blend was adjusted to illuminate a circular area with a radius of 58 μm. The 580-nm LED was manually turned on after 1 min of imaging and off after 4 min of imaging. A Prior XY stage (Prior Scientific, Cambridge, UK) was manually operated to keep specifically the tail of the worm in the by the LED-illuminated area during stimulation. Worms were imaged with a frame rate of 8 Hz. Only every eighth image was used for analysis.

### Activity measurements of command interneurons

GCaMP3.3 was expressed in command interneurons using the *glr-1* promoter [100]. L1 larvae were placed in microfluidic chambers and were imaged using a time-lapse protocol. One DIC and one GFP image was taken every 8 s using a 20× objective and a 0.7 lens. The 490-nm intensity for GFP imaging was set to 0.15 mW/mm$^2$. Intensity values of all command interneurons located in the head of worms were extracted manually and analyzed as one entity.

### Pan-neuronal activity measurements

GCaMP6s and RFP were pan-neuronally expressed under the *rab-3* promoter [101]. As in the activity measurements of command interneurons, L1 lethargus was imaged in microfluidic devices. For the optogenetic experiment (Fig 6C), every 30 min, 20 DICs were taken first in order to determine lethargus. This was followed by GFP images that were taken all 5.8 s for 9 min. The 490-nm intensity was set to 0.07 mW/mm$^2$. In the blue light stimulation experiment, additional RFP images were taken. A custom-written MATLAB code detected the mean intensity of all head neurons in each GFP and RFP frame. The head neurons were thus analyzed as one entity.

### Blue-light stimulation experiments

L1 worms were placed in microfluidic chambers for blue light stimulation experiments. The protocol was repeated every 15 min. First, 20 DIC pictures were taken every 500 ms to determine whether the worm was pumping or not. Next, baseline GCaMP was imaged for 3 min, the stimulation phase then lasted 18 s, and a recovery phase was imaged for 3 min. The 490-nm intensity for calcium imaging was 0.07 mW/mm$^2$. The 490-nm intensity for stimulation was set to 1.01 mW/mm$^2$ with a 20× objective. The same LED was used for calcium imaging and stimulation. The intensity levels were controlled with Andor IQ2 software.

The RFP signal of the pan-neuronal strain was imaged in addition to the GCaMP signal during the protocol every 3 s with 585-nm LED illumination, which was set to 0.17 mW/mm$^2$.

### Mechanical stimulation using dish tapping

The mechanical tapping set up was described previously [67,102]. L1 larvae were imaged in microfluidic chambers using a 20× objective and a 0.7 lens. Microfluidic chambers were put in

a specialized dish. The dish was tapped by a piston driven by an electromagnet. The piston and the electromagnet were held in a homemade aluminum frame as described previously [102] (model used was Kuhnke, product number H2246). The voltage used for stimulus application was 5 V; the tapping stimulus was applied between image acquisition using TTL triggering to avoid blurring. Imaging was controlled with Andor IQ2 software. The imaging protocol was repeated every 15 min. First, 20 DIC pictures were taken with a frequency of 2 pictures per second to determine the status of worms. Throughout all following steps, GCaMP measurements were taken every 3 s. The 490-nm intensity for calcium imaging was 0.15 mW/mm$^2$. Baseline GCaMP was measured over 3 min. Following the tap, GCaMP was measured for 3 min. This experiment was initially planned to be combined with optogenetic stimulation, therefore a weak green light stimulus was applied, starting 15 s before and ending 45 s after the tapping stimulation. The part of the experiment during which a green light stimulus was applied was selected for presentation in this study. Green light (585 nm) for stimulation was set to 0.17 mW/mm$^2$. Because we did not see any noticeable changes upon applying green light, we presume that it does not strongly affect the experiment.

### Simultaneous calcium imaging of RIS and PVC

In order to simultaneously image RIS and PVC, L1 lethargus worms were transferred from a growing plate using a platinum wire worm pick and were fixed on 10% agarose pads with 0.1 μm Polybead microspheres [99] and 25 mM Levamisole. The worms were then imaged through a 40× oil objective with an image taken every 3 s for 30 min with 490-nm light of 1.35 mW/mm to image GCaMP. Fluorescence intensities for PVC and RIS were cropped by using a region of interest. A custom-written MATLAB script then detected all RIS peaks. For this, the GCaMP data were first smoothed over 30 values through the in-built function "smooth," which is a first-degree polynomial local regression model. Through the in-built MATLAB function "islocalmax," and a minimum prominence value of 0.2, the locations of RIS peaks were detected. The data of RIS as well as PVC GCaMP intensity were aligned to the detected RIS peak location.

### Spinning disc confocal microscopy

L4 worms were fixed with Levamisole. Spinning disc imaging was done with an Andor Revolution disc system (Andor Technology Ltd.) using a 488-nm (0.34 mW/mm$^2$) and a 565-nm (0.34 mW/mm$^2$) laser and a Yokogawa (Japan) CSU-X1 spinning disc head. Worms were imaged through a 100× oil objective. In S6A and S6B Fig, an additional 1.5 lens was used. z-Stacks with z-planes 0.5 μm apart spanning a total distance of 10 μm were taken, and a maximum intensity projection was calculated in ImageJ (developed by Wayne Rasband, open source).

### Tail-touch experiment

L4 worms were grown and filmed on NGM plates with OP50 bacteria at 20 ˚C. An eyelash was used to gently touch the tail of the worms during L4 lethargus. The time from tail touch until the worms were immobile again was measured with a timer. If worms did not mobilize upon tail touching, the time was counted as zero. For GCaMP intensities, worms were imaged before and after tail touch each second for a total of 30 s. They were illuminated with a Leica EL6000 LED (Leica, Wetzlar, Germany).

## Image analysis

Image sequences for analysis were selected either based on lethargus or molting time points. Lethargus was determined through DIC or bright-field images as the nonpumping phase before molting. Time points were classified to be in or outside of lethargus. Typically, the entire lethargus time and 2 h before lethargus were analyzed. Worms that were immobilized during the measurements were classified according to their pumping behavior on NGM plates directly before imaging. Two parameters were extracted from the image sequences, as follows.

1. Calcium signals were extracted automatically or manually with custom-written MATLAB codes. These codes extracted defined regions of each image and detected intensity and position data. Extracted regions were chosen slightly bigger than the sizes of measured neurons. From these extracted regions, a certain percentage of highest-intensity pixel was taken as signal. The remaining pixels were taken as background. From the signal, the background was subtracted. For the pan-neuronal and interneuron activity measurements, the signal in the head was treated as one large neuron and analyzed in the same way as single neurons. All head neurons expressed under the *rab-3* promoter were included in the pan-neuronal GCaMP measurements.

   For all stimulation experiments (optogenetic and blue light stimulation experiments), the baseline measurement of each time point was utilized for signal normalization and $\Delta F/F$ generation, except for Fig 6C. In Fig 6C, a mean of all baseline intensities for all wake time points for each worm was calculated. The mean was then utilized for normalization for all time points for each worm to better show the different RIS activities during wake and sleep. The pan-neuronal signal in Fig 6B was normalized over the measured RFP signal to retrieve $\Delta R/R$. For the transient alignments in Fig 3G, peaks and corresponding speeds were extracted through a custom-written MATLAB script and aligned as time point zero.

2. The speeds of the worms were calculated from the positions of the tracked neuron, except for experiments in which no GCaMP intensity was measured. To analyze these experiments, frame subtraction of DIC or bright-field images was done with a custom-written MATLAB routine instead.

## Baseline extraction

In S12A and S12C Fig, the baseline of RIM GCaMP data was extracted by excluding the 95th- to 100th-percentile range for wild type and by excluding the 75th- to 100th-percentile range for *aptf-1(gk794)* mutants. The baseline was smoothed through a second-degree polynomial local regression model and with weighted linear least squares. Zero weight was assigned to data points 6 means outside the absolute deviation. The number of data points used for smoothing was 3%.

## Sleep-bout analysis

Sleep bouts were extracted from selected parts of the time-lapse movies. Dependent on the experiment, a specific period of the movie sequence was selected and processed:

1. The lethargus period (Figs 3F–3J, 4A, 5A, 7, S9A, S9C, S10E–S10I and S12 Figs)

2. The period from 2 h before lethargus up to the end of lethargus (Fig 1A)

3. Either 3 h (Fig 4B–4D and S8C–S8F Fig) or 4 h (S10A–S10D Fig) before shedding of the cuticle

To extract sleep bouts, speeds and subtraction values were first smoothed. In Figs 1A, 3G–3J, 4A–4D, 5A, S8C–S8F, S10, and S12 Figs, speeds were smoothed through a first-degree polynomial local regression model over 20 time points. Other experiments were smoothed through a second-degree polynomial local regression model and with weighted linear least squares. Zero weight was assigned to data points 6 means outside the absolute deviation. Data were smoothed either over 3% of all data (Figs 3H–3J, 7 and S9C–S9F Fig) or over 40 data points (S9A, S9C, S12A and S12C Figs). This was achieved with the "smooth" function in MATLAB. Smoothed speeds were normalized between 0 and 1, with 0 representing the lowest and 1 the highest smoothed speed value of each worm. In order to be scored as a sleep bout, the normalized speed had to be under a defined percentage threshold of the normalized speed for a minimum time. The exact speed and time thresholds were adjusted empirically to represent the worms' behavior [103]. In Fig 3G–3I and S8C–S8F Fig, worms had to have a speed below 5% of their maximum smoothed speeds for at least 2 min in order to be counted as sleeping. For all other experiments, the speed threshold was 10%, and the time threshold was 2 min. The 2-min time threshold was implemented to exclude short pauses of the worm that may not represent sleep bouts. It was determined empirically. The sleep-bout analysis was carried out with a custom-written MATLAB script.

For stimulation experiments, the baseline and recovery time measurements were too short to include a minimum time threshold in the sleep-bout analysis. Therefore, immobility was used as a proxy for sleep. A mean of the wake speeds was calculated for each worm. Depending on the strain used, the worms were counted as sleeping when they were below a threshold of 5% to 30% of their mean wake speed. In most experiments, a worm was counted as sleeping when its speed was below 10% of the calculated mean of the wake speeds. To account for different locomotor behavior of the worms, in S11B Fig, the threshold was adjusted to 5%; in S8G Fig, to 20%; in S7B Fig, to 25%; in Fig 6C, S1C, S2A–S2C, S3, S5B, S9D, S9E, and S11I Figs, to 30%; and in S11C–S11E Fig, to 50%. RIS signals and speeds of wild type and mutants were aligned to sleep-bout onset for comparison in Figs 3H and 4A, S8F, S9A, S10D and S10H Figs. RIM signals and speeds were aligned to sleep-bout onset in Fig 4A. For GCaMP normalization, 10 data points before sleep-bout onset were taken as baseline in order to calculate ΔF/F. In S11A Fig, motion bouts were assigned whenever there was no detected sleep bout.

## RIS peak alignment

For RIS wide peak detection (Fig 3G–3H), first the normalized GCaMP data were smoothed over 60 time points with the in-built MATLAB function "smooth." Wide peaks were then detected with the in-built MATLAB function "findpeaks" and a minimum peak prominence threshold of 0.15. GCaMP intensities, speeds, and sleep fractions were then aligned to the detected peak maxima. Analysis for narrow peaks was conducted similarly; only 2 aspects were changed (S8C–S8F Fig). To find narrow peaks, smoothing was limited to only 5 time points, and a minimum peak prominence threshold was set to 0.2.

## Detection of direction of movement

The direction of movement was analyzed with a custom-written MATLAB script. This MATLAB script took 2 points, the nose and the pharynx, to calculate the direction. For 2 consecutive images, the distance of the nose in the first image to the pharynx in the first image was compared to the distance of the nose in the second image to the pharynx of the first image. If

the distance increased, the worm was counted as moving forward; if it decreased, it was counted as moving in reverse. If the worm was below a threshold of 2 μm/s, it was counted as sleeping in experiments Figs 2D and 4E. The position of nose and pharynx were detected manually (Figs 2D and 4E). For correction of the stage movement while manually tracking PVC (Fig 2D), the position of a corner of the stage was used.

### Fitting

The data in Fig 6D were fitted to an asymptote, and the data in Fig 6E were fitted to a BoxLucasFit1 with Origin software. The data in S11A Fig were fitted to a logistic regression using Origin software (OriginLab Corporation, Northampton, MA). Exact functions and $R^2$ values can be found in the respective Figures.

### Statistics

Sample sizes were determined empirically based on previous studies. If possible, experiments were carried out with internal controls. If this was not possible, control and experimental condition were alternated. Researchers were not blinded to the genotype for data analysis, as data analysis was performed by automated routines. Sample exclusion is described in the respective Methods sections. To compare GCaMP intensities and speeds of one sample group at different time points, the Wilcoxon signed rank test was utilized. The Fisher's exact test was used to compare the sleep fractions of one sample group at different time points. The entirety of the baseline was compared to the entirety of the stimulation period unless otherwise stated through significance bars. Data from different strains were compared with either the Kolmogorov-Smirnov test or the Welch test. The $p$-values can be taken from the respective Figure descriptions. Depicted in the graph is the mean ± SEM unless otherwise stated. The box in the box plots represents the interquartile range with the median. The whiskers show the 10th- to 90th-percentile range, and the individual data points are plotted on top of the box.

### Supporting information

**S1 Fig. Weak optogenetic RIM depolarization using the *gcy-13* promoter can induce RIS activation or inhibition.** (A) Control experiments. Optogenetic depolarization of RIS presynaptic neurons without the addition of ATR. For statistical calculations, baseline neural activities (0–0.95 min) were compared to neural activity levels during the stimulation period (1–1.95 min). $^*p < 0.05$, $^{**}p < 0.01$, Wilcoxon signed rank test for GCaMP (S2 Data, Sheet S1A). (B) Optogenetic RIC depolarization induced an RIS activity increase outside of and during lethargus. An average of all responsive trials is shown in this figure. Trials were classified as responsive or nonresponsive. In responsive trials, an RIS activity increase correlated with the onset of the stimulation period. In nonresponsive trials, no change in RIS activity levels could be seen. "n" represents the number of animals tested, and "r" represents the number of trials. For statistical analysis, RIS baseline activity levels (0–0.95 min) were compared to activity levels during (1–1.95 min) and after (2–2.95 min) the stimulation. $^*p < 0.05$, $^{**}p < 0.01$, $^{***}p < 0.001$, Wilcoxon signed rank test for GCaMP and speed, Fisher's exact test for sleep fraction (S2 Data, Sheet S1B). (C) Depolarization of RIM using ReaChR expressed under the *gcy-13* promoter had no net effect on RIS function. Neural baseline activity levels (0–0.95 min) were compared to neuronal levels during the stimulation (1–1.95 min) and after the stimulation (2–2.95 min). $^*p < 0.05$, $^{**}p < 0.01$, $^{***}p < 0.001$, Wilcoxon signed rank test for GCaMP and speed, Fisher's exact test for sleep fraction (S2 Data, Sheet S1C-E). (D) RIM optogenetic depolarization using ReaChR expressed under the *gcy-13* promoter induced either RIS activation or inhibition. Single trials were classified as activating if an activity increase in RIS

correlated with onsets of optogenetic stimulation periods. Trials were classified as inhibitory if an activity decrease in RIS correlated with onsets of optogenetic stimulation periods. "n" represents the number of animals tested, and "r" represents the number of trials. For statistical testing, baseline neural activities (0–0.95 min) were compared to neural activity levels during the stimulation period (1–1.55 min). $^*p < 0.05$, $^{**}p < 0.01$, $^{***}p < 0.001$, Wilcoxon signed rank test for GCaMP and speed, Fisher's exact test for sleep fraction (S2 Data, Sheet S1C-E). (E) Percentage of RIS activation and inhibition following optogenetic RIM activation in different lethargus phases. Lethargus of each individual worm was split into 3 phases of comparable size (lethargus onset, middle of lethargus, and lethargus end). In each interval, for all worms tested the amount of trials showing an RIS activation or RIS inhibition were compared to the total amount of trials in this interval (S2 Data, Sheet S1C-E).
(TIF)

**S2 Fig. RIM inhibition of RIS requires tyramine and FLP-18.** Optogenetic RIM manipulations in these experiments were all performed with ReaChR expressed from the *tdc-1* promoter. (A) Optogenetic RIM depolarization in *flp-18(db99)* single mutants. Outside of lethargus, RIS inactivation caused by RIM optogenetic depolarization was reduced to 37% of wild-type inhibition levels. During lethargus in *flp-18(db99)* mutants, animal inhibition levels were only 25% of wild-type level. Neuronal activity levels before (0–0.95 min), during (1–1.95 min), and after (2.5–2.95 min) optogenetic RIM depolarization were compared. $^*p < 0.05$, $^{**}p < 0.01$, $^{***}p < 0.001$, Wilcoxon signed rank test for GCaMP and speed, Fisher's exact test for sleep fraction (S2 Data, Sheet S2A). (B) Optogenetic RIM depolarization in *tdc-1(n3420)* single mutants. Outside of lethargus, optogenetic RIM depolarization in *tdc-1(n3420)* single mutants no longer induced changes in RIS activity levels. During lethargus, inhibition levels during the stimulation period only reached 40% of wild-type levels. Neuronal activity levels before (0–0.95 min), during (1–1.95 min), and after (2.5–2.95 min) optogenetic RIM depolarization were compared. $^*p < 0.05$, $^{**}p < 0.01$, $^{***}p < 0.001$, Wilcoxon signed rank test for GCaMP and speed, Fisher's exact test for sleep fraction (S2 Data, Sheet S2B). (C) Optogenetic RIM depolarization in *flp-18(db99)* and *tdc-1(n3420)* double mutants had no effect on RIS function. Neuronal activity levels before (0–0.95 min), during (1–1.95 min), and after (2.5–2.95 min) optogenetic RIM depolarization were compared. $^*p < 0.05$, $^{**}p < 0.01$, $^{***}p < 0.001$, Wilcoxon signed rank test for GCaMP and speed, Fisher's exact test for sleep fraction (S2 Data, Sheet S2C). (D) Quantification of inhibition strength. RIS activity levels during optogenetic RIM depolarization in *flp-18(db99)*, *tdc-1(n3420)* and *flp-18(db99)*, and *tdc-1(n3420)* double mutants were compared to wild-type levels. Wild-type data are depicted in Fig 1B, RIM panel. Inhibition strength was calculated by subtracting RIS activity levels before the stimulation (0–0.95 min) from activity levels during the stimulation (1–1.95 min). Samples were tested for normal distribution using the Shapiro-Wilk test. Wild type and mutants were compared with a Welch test. $^{***}p < 0.001$ (S2 Data, Sheet S2D-E). (E) Quantification of RIS activity levels following RIM optogenetic depolarization. Activity levels in *flp-18(db99)*, *tdc-1(n3420)* and *flp-18(db99)*, and *tdc-1(n3420)* double mutants were compared to wild-type levels. Wild-type data are depicted in Fig 1B in the RIM panel. For statistical calculations, RIS activity levels before the stimulation (0–0.95 min) were subtracted from activity levels after the stimulation (2.5–2.95 min). Samples were tested for a normal distribution using the Saphiro-Wilk test. To compare genotypes, a Welch test was performed for all conditions, except for the comparison of activity levels between wild type and *tdc-1(n3420)* single mutants during lethargus. The *tdc-1(n3420)* data were not normally distributed, and thus a Kolmogorov-Smirnov test was used. $^{***}p < 0.001$ (S2 Data, Sheet S2D-E).
(TIF)

**S3 Fig. RIM activation of RIS requires glutamatergic signaling.** (A) RIM optogenetic depolarization using ReaChR expressed under the *gcy-13* promoter induced robust RIS activation in L4 larvae. In the L4 larvae, RIS activation by RIM optogenetic depolarization was more robust compared with the same experiment in L1 larvae. No trial selection was required. For statistical analysis, RIS baseline activity levels (0–0.95 min) were compared to activity levels during (1–1.95 min) and after (2–2.95 min) the stimulation. $^{*}p < 0.05$, $^{**}p < 0.01$, $^{***}p < 0.001$, Wilcoxon signed rank test for GCaMP and speed, Fisher's exact test for sleep fraction (S2 Data, Sheet S3A). (B) The activating input of RIM optogenetic depolarization on RIS was almost completely abolished in *eat-4(ky5)* mutants. For statistical analysis, RIS baseline activity levels (0–0.95 min) were compared to activity levels during (1–1.95 min) and after (2–2.95 min) the stimulation. $^{*}p < 0.05$, $^{**}p < 0.01$, $^{***}p < 0.001$, Wilcoxon signed rank test for GCaMP and speed, Fisher's exact test for sleep fraction (S2 Data, Sheet S3B).
(TIF)

**S4 Fig. Activation of RIS by PVC is strongly enhanced during lethargus.** Optogenetic PVC depolarization in L2 larvae led to RIS activation outside of and during lethargus. RIS activation during lethargus was strongly enhanced. Plotted data represent the average over all experimental trials. Neural activity levels before the stimulation (0–0.95 min) were compared to activity levels during the stimulation (1–1.95 min). $^{*}p < 0.05$, $^{**}p < 0.01$, Wilcoxon signed rank test (S2 Data, Sheet S4).
(TIF)

**S5 Fig. Optogenetic hyperpolarization experiments.** (A) Control experiments. Optogenetic hyperpolarization of RIS presynaptic neurons without the addition of ATR. For statistical calculations, baseline neural activities (0–0.95 min) were compared to neural activity levels during the stimulation period (1–1.95 min). $^{*}p < 0.05$, $^{**}p < 0.01$, Wilcoxon signed rank test for GCaMP (S2 Data, Sheet S5A). (B) Hyperpolarization of RIM using ArchT expressed under the *gcy-13* promoter had no net effect on RIS function. Neural baseline activity levels (0–0.95 min) were compared to neuronal levels during the stimulation (1–1.95 min) and after the stimulation (2–2.95 min). $^{*}p < 0.05$, $^{**}p < 0.01$, $^{***}p < 0.001$, Wilcoxon signed rank test for GCaMP and speed, Fisher's exact test for sleep fraction (S2 Data, Sheet S5B, C). (C) RIM optogenetic hyperpolarization using ArchT expressed under the *gcy-13* promoter caused a decrease in RIS activity levels in selected trials. Single trials were classified as activating if an activity increase in RIS occurred at the onset of the optogenetic stimulation period. Trials were classified as inhibitory if an activity decrease in RIS occurred at the onset of the optogenetic stimulation period. "n" represents the number of animals tested, and "r" represents the number of trials. For statistical calculations, neural baseline activity levels (0–0.95 min) were compared to levels during the stimulation period (1–1.75 min). $^{*}p < 0.05$, $^{**}p < 0.01$, $^{***}p < 0.001$, Wilcoxon signed rank test for GCaMP and speed, Fisher's exact test for sleep fraction (S2 Data, Sheet S5B, C). (D) Simultaneous optogenetic hyperpolarization of CEP and URY neurons does not induce changes in RIS activity levels. For statistical testing, baseline neural activities (2–2.95 min) were compared to neural activity levels during the stimulation period (3–3.95 min) and after the stimulation (6–6.95 min). $^{**}p < 0.01$, $^{***}p < 0.001$, Wilcoxon signed rank test for GCaMP and speed, Fisher's exact test for sleep fraction (S2 Data, Sheet S5D).
(TIF)

**S6 Fig. *zk673.11* is expressed in PVC, RID, and cholinergic motor neurons.** (A–B) Expression of *nmr-1* and *zk673.11* only overlaps in PVC in the tail. (C–D) Expression of *nmr-1* and *zk673.11* does not overlap in head neurons.
(TIF)

**S7 Fig. PVC has multiple functions.** (A) PVC excitability remained unchanged during lethargus. Experiments were performed in immobilized L1 larvae to ensure PVC-specific green light illumination. A long baseline of 10 min was used to achieve stable baseline conditions. Activity levels of PVC during optogenetic depolarization were indistinguishable outside and during lethargus. PVC displayed a negative rebound transient after optogenetic depolarization. However, there was no difference in the amount of negative rebound outside and during lethargus (S2 Data, Sheet S7A). (B) RIS showed a rebound after mechanical stimulation. This rebound was stronger in worms during lethargus, and only during lethargus was the RIS rebound accompanied by a strongly increased immobilization of worms. $^{**}p < 0.01$, $^{***}p < 0.001$, Wilcoxon signed rank test for GCaMP and speed, Fisher's exact test for sleep fraction (S2 Data, Sheet S7B). (C–D) Effects of PVC stimulation on AVB activity. L1 larvae were immobilized for optogenetic experiments to ensure cell-specific stimulation of PVC. AVB activated upon optogenetic PVC depolarization with the same response strength during and outside of lethargus. AVB displayed an oscillatory activity pattern in 44% of all trials in worms outside of lethargus. AVB activity oscillated in 70% of all trials during lethargus. $^{*}p < 0.05$, $^{**}p < 0.01$, Wilcoxon signed rank test for GCaMP (S2 Data, Sheet S7C-D).
(TIF)

**S8 Fig. Effects of optogenetic RIS activation and inhibition on PVC and RIM activity.** (A) RIS depolarizes during optogenetic activation in fixed animals. As controls, experiments were performed in the absence of ATR. $^{***}p < 0.001$, Wilcoxon signed rank test (S2 Data, Sheet S8A). (B) RIS hyperpolarization led to a weak PVC depolarization outside and during lethargus. For statistical calculations, neural activities before the stimulation period (0–1 min) were compared to activity levels during the stimulation period (1–2 min). $^{*}p < 0.05$, $^{**}p < 0.01$, compared before and during stimulation, Wilcoxon signed rank test (S2 Data, Sheet S8B). (C–D) Sample trace of RIS activity and worm locomotion behavior 3 h before shedding of the cuticle of *aptf-1(gk794)* and *flp-11(tm2706)* mutants (S2 Data, Sheet S8C and S8D). (E–F) *flp-11 (tm2706)* mutants have a significantly increased number of short RIS peaks that do not correlate with sleep. (E) $^{**}p < 0.01$, $^{***}p < 0.001$, Welch test. (F) $^{**}p < 0.01$, Kolmogorov-Smirnov test (S1 Data, Sheet 3G-I). (G) Optogenetic RIS depolarization has no effect on RIM activity outside of and during lethargus. Neuronal activity levels before (0–0.95 min) and during (1–1.95 min) the stimulation period were compared. $^{*}p < 0.05$, $^{**}p < 0.01$, $^{***}p < 0.001$, Wilcoxon signed rank test for GCaMP and speed, Fisher's exact test for sleep fraction (S2 Data, Sheet S8G). (H) Optogenetic RIS hyperpolarization induced increased RIM activity both outside of and during lethargus. Measurements were performed in immobilized L1 larvae to reduce measurement noise. Activity levels during baseline measurements (0–0.95 min) were compared to levels during optogenetic RIS manipulation (1–1.95 min). $^{*}p < 0.05$, Wilcoxon signed rank test for GCaMP (S2 Data, Sheet S8H).
(TIF)

**S9 Fig. Command interneurons are required for RIS activation and sleep induction.** (A) RIS activation in sleep bouts was strongly reduced in command-interneuron–ablated worms. Samples were tested for normal distribution using the Saphiro-Wilk test. $^{*}p < 0.05$, Welch test (S2 Data, Sheet S9A-C). (B) Command-interneuron–ablated worms moved much slower than wild-type worms. Command interneurons were genetically ablated by expressing ICE from the *nmr-1* promoter. Samples were tested for normal distribution using the Saphiro-Wilk test. $^{***}p < 0.001$, Welch test for the wake condition and Kolmogorov-Smirnov test for the sleep condition (S2 Data, Sheet S9A-C). (C) Sample traces of RIS activity levels and worm locomotion behaviors outside of and during lethargus in command-interneuron–ablated worms and wild-type worms. In command-interneuron–ablated worms, quiescence bouts occurred only

around the middle of the lethargus period (S2 Data, Sheet S9A-C). (D–E) Mosaic analysis of worms expressing an extrachromosomal array of *nmr-1::ArchT*. Worms were selected that expressed the transgene only in head neurons (D) or head neurons and PVC (E). Neuronal activity levels before (2–2.95 min) and during (3–3.95 min) the stimulation period was compared. $*p < 0.05$, $**p < 0.01$, $***p < 0.001$, Wilcoxon signed rank test for GCaMP and speed, Fisher's exact test for sleep fraction (S2 Data, Sheet S9D and S9E).
(TIF)

**S10 Fig. Glutamatergic signaling is required for sleep induction.** (A–D) Sleep-bout analysis of *eat-4(ky5)* mutant larvae. *eat-4(ky5)* animals lacked significant RIS activation at sleep-bout onset. Consistent with this finding, mutant worms displayed a strong reduction in quiescence during lethargus. Samples were tested for a normal distribution using the Saphiro-Wilk test. $**p < 0.01$, $***p < 0.001$, Welch test for comparisons of sleep-bout lengths, sleep-bout frequencies, and sleep fractions. Wilcoxon signed rank test for quantifications of RIS activity levels in sleep bouts (S2 Data, Sheet S10A-D). (E–I) Sleep-bout analysis of *nmr-1(ak4)* mutant animals. RIS activity levels in sleep bouts were slightly reduced in the mutant. *nmr-1(ak4)* mutants did not show a reduced amount of quiescence during lethargus. Samples were tested for a normal distribution using the Saphiro-Wilk test. $*p < 0.05$, Welch test for comparisons of sleep-bout frequencies, sleep fractions, maximum RIS activity levels in sleep bouts, and RIS activity levels at the end of sleep bouts. Kolmogorov-Smirnov test for the comparison of sleep-bout lengths (S2 Data, Sheet S10E-I).
(TIF)

**S11 Fig. RIS rebound activation following optogenetic hyperpolarization requires synaptic transmission.** (A) RIS GCaMP transient intensities in wild-type worms are correlated with the length of the preceding motion bout. The longer the preceding motion bout, the stronger the RIS activation (S2 Data, Sheet S11A). (B) RIM was inhibited during and post hyperpolarization. $*p < 0.05$, $**p < 0.01$, $***p < 0.001$, Wilcoxon signed rank test for GCaMP and speed, Fisher's exact test for sleep fraction (S2 Data, Sheet S11B). (C) RIS was optogenetically hyperpolarized with stimuli lasting for 48 s (C), 5 min (D), or 10 min (E). Worms not showing a rebound activation transient were excluded from the analysis, which was no worm for 48 s-, 1 out of 7 worms for 5 min-, and 1 out of 13 worms for 10-min stimulation experiments. Data from these plots were used to generate a dose-response curve of optogenetic RIS hyperpolarization (Fig 6D and 6E). $*p < 0.05$, $**p < 0.01$, $***p < 0.001$, Wilcoxon signed rank test for GCaMP and speed, Fisher's exact test for sleep fraction (S1 Data, Sheet 6D,E). (F–H) Following optogenetic hyperpolarization, RIS displayed strong rebound activation during lethargus (F). Rebound activation was abolished in a strain that is deficient for neurotransmission specifically in RIS (*flp-11::TetX*). (G) Rebound activation was abolished also by a mutation that impaired global synaptic transmission (*unc-13(s69)*). (H) $*p < 0.05$, $**p < 0.01$, Wilcoxon signed rank test (S2 Data, Sheet S11F-H). (I) RIS showed a negative rebound following its own optogenetic depolarization. The strength of the negative rebound transient depended on the lethargus status of the worm. Worms during lethargus displayed a 3-times-stronger negative rebound compared to worms outside of lethargus. $*p < 0.05$, $**p < 0.01$, $***p < 0.001$, Wilcoxon signed rank test for GCaMP and speed, Fisher's exact test for sleep fraction (S2 Data, Sheet S11I).
(TIF)

**S12 Fig. RIM baseline activity levels are dampened during lethargus independently of RIS.** (A) Sample traces of RIM transient frequencies, RIM baseline activities, and worm locomotion behaviors outside of and during lethargus in wild-type worms and *aptf-1(gk794)* mutants (S2

Data, Sheet S12). (B) Wild-type worms, but not *aptf-1(gk794)* mutant worms, display changes in RIM transient frequencies across lethargus. Transient frequencies were assessed manually. To be counted as a transient, RIM activity levels had to be at least twice as high as baseline activity levels. ***$p < 0.001$ Kolmogorov-Smirnov test for wild-type condition, Welch test for mutant condition (S2 Data, Sheet S12). (C) The reduction of RIM baseline activity levels during lethargus is preserved in *aptf-1(gk794)* mutants. **$p < 0.01$, Wilcoxon signed rank test (S2 Data, Sheet S12).
(TIF)

**S13 Fig. Assaying gentle tail touch reveals an inhibitory role of RIM on RIS.** (A) RIM ablation increases the reinstating of immobility following gentle tail touch during lethargus. *$p < 0.05$, Kolmogorov-Smirnov test (S2 Data, Sheet S13A). (B) RIM ablation increases RIS activation in response to gentle tail touch. **$p < 0.01$. Kolmogorov-Smirnov test (S2 Data, Sheet S13B).
(TIF)

**S1 Text. A list of strains that were used during this study.**
(DOCX)

**S2 Text. A list of generated constructs during this study.**
(DOCX)

**S3 Text. Sequence of the strain PHX816, which was generated during this study.**
(DOCX)

**S1 Table. List of primers that were used during this study.**
(DOCX)

**S2 Table. List of plasmids that were used during this study.**
(DOCX)

**S3 Table. Experimental details of all optogenetic experiments conducted during this study.**
(DOCX)

**S1 Data. Raw data for all experiments from the main figures (Figs 1–9).**
(XLSX)

**S2 Data. Raw data for the experiments from the supporting figures (S1–S13 Figs).**
(XLSX)

## Acknowledgments

We are grateful to Jonathan Packer and Robert Waterson for suggesting the use of the *zk673.11* promoter. We thank Cori Bargmann, Andrew M. Leifer, Andres Mariqc, Shai Shaham, Yun Zhang, Manuel Zimmer, and the Caenorhabdis Genetics Center for strains. The strain PHX816 was generated by SunyBiotech. We are grateful to Kaveh Ashrafi, Ithai Rabinowitch, and Yoshinori Tanizawa for plasmids.

## Author Contributions

**Conceptualization:** Henrik Bringmann.

**Data curation:** Elisabeth Maluck, Inka Busack, Judith Besseling, Florentin Masurat.

**Formal analysis:** Elisabeth Maluck, Inka Busack, Judith Besseling, Florentin Masurat.

**Funding acquisition:** Henrik Bringmann.

**Investigation:** Elisabeth Maluck, Inka Busack.

**Methodology:** Elisabeth Maluck, Inka Busack, Judith Besseling, Florentin Masurat.

**Project administration:** Henrik Bringmann.

**Resources:** Elisabeth Maluck, Inka Busack, Judith Besseling, Florentin Masurat, Michal Turek, Karl Emanuel Busch.

**Supervision:** Henrik Bringmann.

**Writing – original draft:** Henrik Bringmann.

**Writing – review & editing:** Elisabeth Maluck, Inka Busack, Karl Emanuel Busch.

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
