## [Editor Report · Decision Letter 0]

19 Jun 2019

Dear Dr Bringmann, 

Thank you for submitting your manuscript entitled "A tripartite flip-flop sleep circuit switches sleep states" for consideration as a Research Article by PLOS Biology.

Your manuscript has now been evaluated by the PLOS Biology editorial staff as well as by an Academic Editor with relevant expertise and I am writing to let you know that we would like to send your submission out for external peer review.

**IMPORTANT**: Please also see below for further information regarding completing the MDAR reporting checklist. The checklist can be accessed here: https://plos.io/MDARChecklist

Please re-submit your manuscript and the checklist, within two working days, i.e. by Jun 21 2019 11:59PM.

Kind regards,

Hashi Wijayatilake, PhD,

Managing Editor

PLOS Biology

INFORMATION REGARDING THE REPORTING CHECKLIST:

PLOS Biology is pleased to support the "minimum reporting standards in the life sciences" initiative (https://osf.io/preprints/metaarxiv/9sm4x/). This effort brings together a number of leading journals and reproducibility experts to develop minimum expectations for reporting information about Materials (including data and code), Design, Analysis and Reporting (MDAR) in published papers. We believe broad alignment on these standards will be to the benefit of authors, reviewers, journals and the wider research community and will help drive better practise in publishing reproducible research. 

We are therefore participating in a community pilot involving a small number of life science journals to test the MDAR checklist. The checklist is intended to help authors, reviewers and editors adopt and implement the minimum reporting framework. 

IMPORTANT: We have chosen your manuscript to participate in this trial. The relevant documents can be located here:

MDAR reporting checklist (to be filled in by you): https://plos.io/MDARChecklist

**We strongly encourage you to complete the MDAR reporting checklist and return it to us with your full submission, as described above. We would also be very grateful if you could complete this author survey:

https://forms.gle/seEgCrDtM6GLKFGQA

Additional background information:

Interpreting the MDAR Framework: https://plos.io/MDARFramework

Please note that your completed checklist and survey will be shared with the minimum reporting standards working group. However, the working group will not be provided with access to the manuscript or any other confidential information including author identities, manuscript titles or abstracts. Feedback from this process will be used to consider next steps, which might include revisions to the content of the checklist. Data and materials from this initial trial will be publicly shared in September 2019. Data will only be provided in aggregate form and will not be parsed by individual article or by journal, so as to respect the confidentiality of responses. 

Please treat the checklist and elaboration as confidential as public release is planned for September 2019.

We would be grateful for any feedback you may have.

---

## [Decision Letter · Decision Letter 1]

9 Jul 2019

Dear Dr Bringmann,

Thank you very much for submitting your manuscript "A tripartite flip-flop sleep circuit switches sleep states" for consideration as a Research Article at PLOS Biology. Your manuscript has been evaluated by the PLOS Biology editors, an Academic Editor with relevant expertise, and by several independent reviewers.

In light of the reviews (below), we will not be able to accept the current version of the manuscript, but we would welcome resubmission of a revised version that takes into account the reviewers' comments. Three of the reviewers have requests for some additional experiments to strengthen your conclusions. The Academic Editor agrees these requested experiments will significantly improve the manuscript. Please note that we cannot make any decision about publication until we have seen the revised manuscript and your response to the reviewers' comments. Your revised manuscript is also likely to be sent for further evaluation by the reviewers.

Your revisions should address the specific points made by each reviewer. Please submit a file detailing your responses to the editorial requests and a point-by-point response to all of the reviewers' comments that indicates the changes you have made to the manuscript. In addition to a clean copy of the manuscript, please upload a 'track-changes' version of your manuscript that specifies the edits made. This should be uploaded as a "Related" file type. You should also cite any additional relevant literature that has been published since the original submission and mention any additional citations in your response. 

Before you revise your manuscript, please review the following PLOS policy and formatting requirements checklist PDF: http://journals.plos.org/plosbiology/s/file?id=9411/plos-biology-formatting-checklist.pdf. It is helpful if you format your revision according to our requirements - should your paper subsequently be accepted, this will save time at the acceptance stage.

Please note that as a condition of publication PLOS' data policy (http://journals.plos.org/plosbiology/s/data-availability) requires that you make available all data used to draw the conclusions arrived at in your manuscript. If you have not already done so, you must include any data used in your manuscript either in appropriate repositories, within the body of the manuscript, or as supporting information (N.B. this includes any numerical values that were used to generate graphs, histograms etc.). For an example see here: http://www.plosbiology.org/article/info%3Adoi%2F10.1371%2Fjournal.pbio.1001908#s5.

For manuscripts submitted on or after 1st July 2019, we require the original, uncropped and minimally adjusted images supporting all blot and gel results reported in an article's figures or Supporting Information files. We will require these files before a manuscript can be accepted so please prepare them now, if you have not already uploaded them. Please carefully read our guidelines for how to prepare and upload this data: https://journals.plos.org/plosbiology/s/figures#loc-blot-and-gel-reporting-requirements.

We expect to receive your revised manuscript within two months. Please email us (plosbiology@plos.org) to discuss this if you have any questions or concerns, or would like to request an extension. At this stage, your manuscript remains formally under active consideration at our journal; please notify us by email if you do not wish to submit a revision and instead wish to pursue publication elsewhere, so that we may end consideration of the manuscript at PLOS Biology.

When you are ready to submit a revised version of your manuscript, please go to https://www.editorialmanager.com/pbiology/ and log in as an Author. Click the link labelled 'Submissions Needing Revision' where you will find your submission record. 

Sincerely,

Hashi Wijayatilake, PhD, 

Managing Editor

PLOS Biology

REVIEWS:

Reviewer #1: 

I think this manuscript is a well conducted study that employs a series of circuit manipulations that nicely demonstrate the role of the PVC neuron in activating the sleep active RIS neuron in C. elegans. 

The authors give a thorough introduction to the question they are trying to elucidate. Here, this reviewer really believes that the use of such a powerful model as C.elegans with its stereotype and complete connectome is of great benefit to the question at hand. This gives a lot of credit to the study.

 This reviewer believes that the conclusions made by the authors are supported by the data and more importantly that the paper manages to provide a reasonably clear explanation of the complex interactions occurring between the different types of neurons assayed in this study.

This reviewer has no major concerns regarding this manuscript and believes it could be publish as it is.

--

Reviewer #2: 

A key outstanding question in understanding sleep/wake control is “how is behavioral state stabilized?”, i.e. “Why is there no rapid switching between sleep and wake states?” as would be expected from an electronic flip-flop switch. One proposal is that there is an excitatory input (e.g. orexin neuropeptides in mammals) to wake-active neurons, which stabilizes their activity. Here, Maluk et al propose an alternative (or additional) explanation for behavioral state stability. Based on their circuit-based analysis of sleep in C. elegans, they propose that a pair of neurons (called PVC) that is involved in wake behavior, also forms reciprocal excitatory connections with a sleep inducing neuron (called RIS) to create a reinforcing positive feedback loop to stabilize the sleep state. This model is particularly provocative since it invites a reconsideration of a view held for over 35 years that the main role of PVC neurons is to promote forward locomotion in response to mechanical stimulation of the posterior part of the body. The authors additionally suggest a circuit-based mechanism for the homeostatic regulation of sleep. 

While the results are solid and support their model, there are a few key experiments missing that would shore up the key claims. 

1. The claim that PVC activation is part of the circuit for sleep initiation is supported by their neuronal activation and silencing experiments. However, these optogenetic manipulations are artificial and could possibly generate non-physiological responses. If their model is correct, one would expect to see spontaneous PVC activity increase coincident with RIS activity and during lethargus quiescent bouts. 

2. The authors show that experimental depolarization of RIS leads to a rebound reduced calcium activity whereas hypopolarization of RIS leads to rebound increased calcium activity. However, they did not show whether such apparent homeostatic regulation is unique to RIS or whether it also occurs in PVC or neurons. This is important since neuronal activity and synaptic homeostasis have been widely described in both mammals and invertebrates, even in neurons not involved in sleep regulation. Moreover, an experiment that could distinguish between homeostatic mechanisms intrinsic to RIS and a circuit based mechanism is to repeat these experiments in a genetic background in which synaptic transmission is blocked (e.g. using unc-13 mutants). 

3. Lines 499-501. The authors proposal that the degree of PVC activation determines whether it causes forward locomotion or activates RIS to cause quiescence is testable. It would predict that weak PVC optogenetic stimulation results in forward movement whereas strong stimulation results in quiescence. This would seems to be a simple experiment for them to do with existing reagents. It would allow them to potentially reconcile the apparently conflicting functions of PVC. 

I noted several issues with the text that need corrections. These writing issues and other minor points are listed below.

1. Last sentence in abstract “A tripartite flip-flop switch likely also underlies sleep control in mammals.” They provide no literature support to this claim. I suggest instead changing “…likely also underlies…” to “…may also underlie…”

2. In the abstract, the authors say “PVC is inhibited by reverse command interneurons, which are stimulated by arousing cues.” Yet in their bullet Highlight points you say “Wakefulness-activated PVC forward command interneurons”. It is not clear to the reader what the difference is between “arousing cues” and “wakefulness”.

3. Highlight point “RIS activity is homeostatically regulated in both directions” is unclear. I would suggest revising to “RIS activity shows compensatory homeostatic regulation following its depolarization or hyperpolarization”. 

4. Last highlight point. It is unclear what the authors mean by “neuronal dampening”. Do they mean “developmental time dependent global dampening of neural activity”? 

5. The authors use the term “tiredness” several times but I think they mean “sleepiness”. “Tiredness” is less specific as it can also refer to muscle fatigue and to mental fatigue that does not reflect sleep/wake regulation. 

6. Lines 48-49: “Sleep deprivation is detrimental to human health and is associated with an increased risk of infection, cardiovascular disease, depression, obesity, type 2 diabetes, and cancer”. The authors should be careful about these claims (also they provide no references for this sentence.). A more accurate capture of the known literature would “Short sleep is associated with an increased risk of infection (REFs), cardiovascular disease (REF), depression (REF), obesity (REF), type 2 diabetes (REF), and cancer (REF)”. 

7. Lines 52-56 on aplysia and STG ganglia are irrelevant to this paper. Remove.

8. Lines 61: unclear what they mean by the sentence “The depolarization of sleep-active neurons defines the key properties of sleep behavior”.

9. At some point, the authors need to state that the PVC neurons are a pair (not a single neuron). 

10. Lines 124-125, unclear what "reverse arousal" means. Do they mean "backwards command interneurons and thus inhibit...” ?

11. Can the variability in the RIM activation effects on RIS be explained by timing of stimulation (early vs late lethargus)?

12. The authors use the terms “optogenetic gain-of-function” and “optogenetic loss of function”. These are terms used to describe gene variant effects but are confusing when used to refer to optogenetic neuronal activation and silencing experiments. 

13. The authors show that command interneuron ablation results in reduced sleep but the more important experiment directly relevant to their model is to ablate PVC specifically. It appears that they have a selective promoter for doing such a kill genetically (perhaps combining miniSOG expressed under their PVC+ promoter coupled with exposure of the tail to blue light). 

14. Figure S4 should be discussed in greater detail. It appears that eat-4 (S4E) is required for sleep maintenance (bout duration) but not for sleep onset (bout frequency). This should be mentioned. Also, nmr-1 is minimally required (S4A) suggesting that other glutamate receptors play a role here.

15. Lines 216-220 sentence is long and unclear: “While PVC has previously been shown to promote forward locomotion[45], its inhibition leads to increased mobility implying that the reduction of locomotion after simultaneous ablation of most command interneurons stems from the reverse command interneurons that promote mobility, whereas PVC appears to play a predominant role in dampening motion behaviors through activation of RIS.”

16. Lines 268-271: “These results are consistent with the idea that sleep induction is a self-enforcing process in which RIS-mediated inhibition of brain activity promotes further RIS activation (Figure 2F).” An alternative explanation for the observations of aptf-1 mutants is that, in this mutant, the RIS neuron is intrinsically defective in more ways than just lacking flp-11. Doing this experiment in flp-11 mutants would be cleaner and more convincing.

17. Lines 34 4-345: “Hyperactive mutants suppress sleep across species including C. elegans [49-51]”. The authors should cite should also cite Nagy et al ELife 2013 and Belfer et al, Sleep 2013 for C. elegans hyper mutants and Cirelli et al Nature 05, Koh et al Science 08, and Kume et al J. Neurosci 05 for Drosophila hyper mutants suppressing sleep. 

18. Line 388: Give reference for tdc-1 promoter.

19. Line 512. I’m not aware of recordings from humans. Do they mean rodents? In this and many other sentences, they do not include a literature reference. 

20. Methods line 584: indicate which of the strains listed were generated during the course of these experiments. For those strains previously described, give reference. 

21. Figure 1CD. Why does RIS activity remain low outside lethargus after PVC inhibition? Similar result in 1D. This bears comment.

22. Figure 1C: How do you know that inhibition was successful in neurons for which the Arch experiments showed no effect on RIS? 

23. Line 895: It was not clear to me why SDQL data was treated differently statistically.

--

Reviewer #3: 

Maluck et al describe a sleep-regulatory circuit in C. elegans that resembles the flip-flop architecture that has been observed in mammals. The authors combine optogenetic manipulation and GCaMP imaging in behaving worms to map the connectivity of sleep regulatory neurons during and outside lethargus. The manuscript is well organized and begins to outline the organization of sleep-influencing neurons upstream of RIS, but the following points should be addressed to strengthen the paper:

1) The roles of PVC in directing locomotion and sleep remain unclear. The authors should more clearly describe how PVC activity might promote both locomotion and sleep. Could PVC have two different modes of activity? Does varying the intensity of optogenetic RIS stimulation have different effects on sleep/locomotion?

2) PVC activation increases RIS calcium levels during lethargus, but has no effect or slightly decreases calcium in RIS outside of lethargus. It is not clear how this might occur – does lethargus alter neuromodulatory levels that might render RIS sensitive to PVC activity? Does PVC excitability change during lethargus?

3) While PVC activation increases GCaMP signal in RIS during lethargus but not outside lethargus, PVC hyperpolarization reduces GCaMP in RIS both during and outside lethargus. The authors should discuss how to reconcile these results.

4) If PVC is activated by RIS, then would PVC responses to RIS activation be ablated in aptf-1 mutants that lack neuropeptidergic output from RIS neurons?

5) Lines 240-243 – It is not clearly described how a sleep-command neuron would promote sleep by being active during both sleep and wake.

6) Is the rebound excitation observed in RIS neurons after optogenetic inhibition unique? Do other neurons also show similar homeostatic excitability after acute inhibition? Similarly, is RIS activation increased following mechanical stimulation during lethargus that delays sleep?

7) The model posed in Figure 6 proposes that sleep regulatory signals are relayed through AVE and AVA, but they are not examined in this manuscript. If AVE and AVA act as the wake active arm of a flip-flop switch, then their activity should be suppressed during lethargus. Would AVE and AVA be redundant, or would silencing one be sufficient to suppress arousal and promote sleep? These questions seem central to the flip-flop model.

--

Reviewer #4: 

In this manuscript, Maluck et al. investigate the network regulating the activity of the RIS neuron. This neuron plays a key role in C. elegans developmentally timed sleep, and thus understanding how its activity is regulated is crucial for the elucidation of sleep in this organism and of the general principles of sleep/wake regulation. Using mainly calcium imaging and optogenetics, they show that the PVC neuron is required for RIS activation during sleep, and that RIS can also upregulate PVC activity. Interestingly, they find that none of the pre-synaptic partners of RIS are inhibitory. Based on these observations they suggest that a positive feedback loop upregulate the activity in both neurons during sleep and act as a flip flop mechanism. They also show that continuous inhibition or activation of RIS is followed by compensatory responses both in terms of neuronal activity and in terms of sleep/wake regulation, suggesting that such responses could be involved in sleep homeostasis on a short time scale. Finally, they provide evidence that awakening by a stimulation of the ASH neuron requires the RIM neuron, which is downstream of ASH and upstream of RIS as well as of AVA, a command inhibitory neuron upstream of PVC. I would have two major comments:

1) The authors provide strong evidence that PVC is necessary and sufficient to activate RIS during sleep. They also show that optogenetic stimulation of RIS activates PVC, raising the idea of a positive feed-back loop between the two neurons. Showing that RIS inhibition reduces PVC activity would provide strong support in favor of this feed-back loop, however experiments inhibiting RIS activity are difficult to interpret because such manipulations simultaneously induce wakefulness. Thus the idea that a positive feed-back loop takes place between the two neurons is mainly based on RIS artificial stimulation and data from the literature. The authors suggest in particular that inhibitory inputs linked to reverse locomotion and arousal are likely a key element of this feed-back loop, because they can be modulated by RIS. Ablating specifically the inhibitory input neuron directly connected to both RIS and PVC, i.e. AVE, stimulating RIS and recording PVC activity would provide experimental evidence for this hypothesis, but is not included in the manuscript. The discussion brings forward a flip-flop mechanism based on command interneurons, however other models are possible. Experiments shown in Figure 1 show that the inhibition of PVC does not provide a full awakening and indicate that besides PVC, other neurons with direct synaptic connection to RIS could modulate its activity. It remain possible that manipulating individually each of the non-PVC neurons is insufficient to affect RIS but that a more coordinated inhibition – or activation - of those neurons could tip the balance towards sleep or wakefulness. 

2) The evidence that RIS depolarization is under homeostatic control is based on optogenetic manipulation. It is unclear whether the rebounds observed are a specific feature of RIS or could occur in any neurons manipulated the same way- would stimulation of PVC or RIM be also followed by a rebound? These complementary experiments should be included to strengthen the conclusion. Secondly the expression of channel rhodopsin may affect the activity of the neuron in a way that is not representative of the normal physiology. Regarding that issue, the data shown in Figure S7 is particularly relevant, since RIS activation is not optogenetically induced. However it is not clear in Figure S7 what is the difference between “lethargus non mobilizing” and “lethargus mobilizing”: The first condition induces a ΔF/F rebound and reduced sleep after blue light, while the second is showing an opposite pattern of responses. The authors should clarify. Finally, in Figure 3C and S6 the worms not showing rebound activation transient after RIS were excluded from the analysis. The authors should provide information on the % of responding animals; this is important information regarding the robustness of the phenomenon. 

Other comments:

-Figure 1: Please comment on the fact that RIS is not activated in all sleep episodes, for example episodes 6 and 7 in panel A.

-Figure 3B: The ΔF/F data shown in the right panel does not seem to match with what is shown in figure 3C. 

-in Figure 3C, the amount of sleep detected is much lower than in the other experiments. Please provide explanations in the legend.

-Methods: there is insufficient detail on how pan-neuronal or command interneuron activity was recorded and analyzed.

-Methods: how was the specificity of the genetic ablations checked?

---

## [Decision Letter · Decision Letter 2]

16 Dec 2019

Dear Dr Bringmann,

Thank you for submitting your revised Research Article entitled "A wake-active locomotion circuit depolarizes a sleep-active neuron to induce sleep" for publication in PLOS Biology. I have now obtained advice from three of the original reviewers and have discussed their comments with the Academic Editor. 

Based on the reviews, we will probably accept this manuscript for publication, assuming that you will modify the manuscript to address the remaining points raised by the Reviewer 4. We will not require addition of the data requested by this reviewer but encourage you to add it, if available.

** Please also make sure to address the specific Data Statement-related requests and other policy-related requests noted at the end of this email.

** We notice that your entire Materials and Methods section is in the supplement. We at PLOS Biology discourage this, and instead request that you move whatever possible to a Methods section within the main manuscript (PLOS articles have to page/space limitations). You may keep the strain details, the primer and plasmid tables, and long sequences as supplemental tables/files (referred to from within the main manuscript text), but the remainder of the text should be included in the Methods section in the main manuscript text. Legends for the relevant supplemental figures, tables and files should be included in the main manuscript.

We expect to receive your revised manuscript within two weeks. Your revisions should address the specific points made by the reviewers, as relevant. In addition to the remaining revisions and before we will be able to formally accept your manuscript and consider it "in press", we also need to ensure that your article conforms to our guidelines. A member of our team will be in touch shortly with a set of requests. As we can't proceed until these requirements are met, your swift response will help prevent delays to publication.

*Copyediting*

*Published Peer Review History*

*Early Version*

*Submitting Your Revision*

Sincerely,

Hashi Wijayatilake, PhD, 

Managing Editor

PLOS Biology

DATA POLICY:

You are aware of the PLOS Data Policy, which requires that all data be made available without restriction: http://journals.plos.org/plosbiology/s/data-availability. For more information, please also see this editorial: http://dx.doi.org/10.1371/journal.pbio.1001797

**Thank you for providing the supplemental files with the underlying data for your figures and supplemental figures. 

IMPORTANT:

>>Please also ensure that figure legends in your manuscript include information on where the underlying data can be found, and ensure your supplemental data file/s has a legend.

>>Please ensure that your Data Statement in the submission system accurately describes where your data can be found. Your Data Statement currently reads as follows:

"All data will be available as part of this manuscript. Original data will be provided via a public repository. Strains will be available through the Caenorhabditis elegans Genetics Center. Strains not accepted by the Caenorhabditis elegans Genetics Center will be available from the authors upon request."

> Please confirm and state that all strains *are* available via a repository. 'Available from authors upon request' is not permitted via our Data Policy. Please make sure all strains are made available via a supplemental file if not via a public repository and state their locations.

REVIEWS:

Reviewer #2 (David Raizen): 

The authors have addressed all issues I raised. Nice work. 

Reviewer #3: 

The authors have added a considerable amount of new data for this revision and have addressed my previous comments. I have no new critiques to raise.

Reviewer #4: 

The authors have thoroughly addressed the reviewer's requests. The new results reach more nuanced conclusions, while still providing important information regarding sleep regulation. I would have a few remaining comments:

-The data presented in Figure 2A shows the correlation between PVC and RIS transients. Showing the activity of PVC with respect to sleep bout onset, as presented for RIM in Figure 4A, would add equally important information regarding the link between PVC and sleep, as well as providing a point of comparison with RIM activity.

-Methods: please provide more information on how was the pan-neuronal GCaMP activity evaluated and which neurons were assessed. 

-Methods: please provide more information on how the time threshold of 2 min was chosen for identifying sleep (p22 line 625).

-Main text, p19, line 573: please correct sentence "… RIM continued showing RIS transients during lethargus… "

---

## [Editor Report · Decision Letter 3]

23 Jan 2020

Dear Dr Bringmann,

On behalf of my colleagues and the Academic Editor, Dr. Paul Shaw, I am pleased to inform you that we will be delighted to publish your Research Article in PLOS Biology. 

Early Version

PRESS 

Kind regards,

Krystal Farmer

Development Editor, 

PLOS Biology

on behalf of

Hashi Wijayatilake,

Managing Editor

PLOS Biology